# The RBR E3 ubiquitin ligase HOIL-1 can ubiquitinate diverse non-protein substrates in vitro

Xiangyi S Wang[1,2], Jenny Jiou[1,2], Anthony Cerra[1,2], Simon A Cobbold[1,2], Marco Jochem[1,2], Ka Hin Toby Mak[3,4], Leo Corcilius[3,4], John Silke[2,5], Richard J Payne[3,4], Ethan D Goddard-Borger[2,6], David Komander[1,2], Bernhard C Lechtenberg[1,2]

HOIL-1 is a RING-between-RING-family E3 ubiquitin ligase and a component of the linear ubiquitin chain assembly complex. Although most E3 ubiquitin ligases conjugate ubiquitin to protein lysine sidechains, HOIL-1 has also been reported to ubiquitinate hydroxyl groups in protein serine and threonine sidechains and glucosaccharides, such as glycogen and its building block maltose, in vitro. However, HOIL-1 substrate specificity is currently poorly defined. Here, we show that HOIL-1 is unable to ubiquitinate lysine but can efficiently ubiquitinate serine and a variety of model and physiologically relevant di- and monosaccharides in vitro. We identify a critical catalytic histidine residue, His510, in the flexible catalytic site of HOIL-1 that enables this O-linked ubiquitination and prohibits ubiquitin discharge onto lysine sidechains. We use HOIL-1's in vitro non-proteinaceous ubiquitination activity to produce preparative amounts of different ubiquitinated saccharides that can be used as tool compounds and standards in the rapidly emerging field of non-proteinaceous ubiquitination. Finally, we report an engineered, constitutively active HOIL-1 variant that simplifies in vitro generation of ubiquitinated saccharides.

## Introduction

Ubiquitination is a post-translational modification characterised by the conjugation of the small, 76–amino acid protein ubiquitin to other proteins. Ubiquitination is mediated by a three-tier cascade of enzymes, the E1 ubiquitin–activating enzyme, the E2 ubiquitin–conjugating enzymes, and the E3 ubiquitin ligases (Hershko & Ciechanover, 1998). Ubiquitin is commonly defined as a post-translational protein modification where the ubiquitin C terminus forms an isopeptide bond with a substrate lysine sidechain (Komander & Rape, 2012). Ubiquitin itself contains seven internal lysine residues, allowing ubiquitin to form different types of polyubiquitin chains, each with different biological functions (Komander & Rape, 2012). Ubiquitin can also form peptide bonds with the N-terminal amine of proteins, including ubiquitin itself (Komander & Rape, 2012). In humans, these M1-linked (or linear) ubiquitin chains are thought to be exclusively formed by the RING-between-RING (RBR) E3 ubiquitin ligase HOIP, the catalytic centre of the linear ubiquitin chain assembly complex (LUBAC) (Kirisako et al, 2006; Gerlach et al, 2011; Ikeda et al, 2011; Tokunaga et al, 2011; Smit et al, 2012; Stieglitz et al, 2012, 2013).

In addition to the canonical ubiquitination on amines, non-canonical protein ubiquitination on the hydroxyl groups of Ser and Thr residues via oxyester bonds (O-linked ubiquitination) or the sulphhydryl group of Cys residues via thioester bonds has also been described (Kelsall, 2022; Squair & Virdee, 2022; Dikic & Schulman, 2023). Over the past years, researchers have identified multiple E3 ubiquitin ligases that can form ubiquitin oxyester linkages. MYCBP2 uses a specific Cys-relay system to transfer ubiquitin to Thr residues (Pao et al, 2018). The RBR E3 ligase HHARI (ARIH1) can ubiquitinate Ser or Thr residues on cyclin E and Nrf1 (Purser et al, 2023; Yoshida et al, 2024). The RBR ligase and LUBAC component HOIL-1, which previously showed enigmatic E3 ligase activity (Stieglitz et al, 2012; Smit et al, 2013), has recently been suggested to ubiquitinate Ser and Thr residues on other LUBAC components, the myddosome and ubiquitin (Kelsall et al, 2019; Fuseya et al, 2020; Rodriguez Carvajal et al, 2021; McCrory et al, 2022).

Since 2021, the array of potential ubiquitin substrates has been expanded to non-proteinaceous biomolecules. Ubiquitin has been found conjugated to bacterial LPS, for example, in salmonella-infected cells (Otten et al, 2021), lipids (Sakamaki et al, 2022), ADP-ribose attached to proteins or nucleic acids (Zhu et al, 2022, 2024), saccharides (Kelsall et al, 2022; Yoshida et al, 2024), single-stranded RNA and DNA (Dearlove et al, 2024), and synthetic small molecules (Li et al, 2024 *Preprint*). One of the E3 ligases implicated

[1]Ubiquitin Signalling Division, The Walter and Eliza Hall Institute of Medical Research, Parkville, Australia    [2]Department of Medical Biology, The University of Melbourne, Parkville, Australia    [3]School of Chemistry, The University of Sydney, Sydney, Australia    [4]Australian Research Council Centre of Excellence for Innovations in Peptide and Protein Science, The University of Sydney, Sydney, Australia    [5]Inflammation Division, The Walter and Eliza Hall Institute of Medical Research, Parkville, Australia    [6]ACRF Chemical Biology Division, The Walter and Eliza Hall Institute of Medical Research, Parkville, Australia

Correspondence: lechtenberg.b@wehi.edu.au

in ubiquitination of glucosaccharides is HOIL-1, which can ubiquitinate glycogen, the main storage form of glucose in humans, and its building block maltose in vitro (Kelsall et al, 2022; Wu et al, 2022; Wang et al, 2023; Xu et al, 2023). The diverse set of reported HOIL-1 activities targeting Lys and Ser/Thr sidechains and glucosaccharides prompted us to investigate HOIL-1's relative activity with these different substrates in a more quantitative approach.

Progress in the ubiquitin field over the past two decades was largely driven by the development of highly specific tools and technologies, ranging from chain type–specific antibodies and affimers (Newton et al, 2008; Matsumoto et al, 2010, 2012; Michel et al, 2017), to antibody-based enrichment strategies (Xu et al, 2010; Meyer & Rape, 2014; Akimov et al, 2018) and selective ubiquitin proteomics (Kirkpatrick et al, 2006; Valkevich et al, 2014; Swatek et al, 2019). However, many of these tools are only suitable for Lys- or protein-conjugated ubiquitin. Therefore, although non-proteinaceous ubiquitination is a rapidly emerging field, progress is currently hampered by the lack of tools and standards to study these novel modifications (Lechtenberg & Komander, 2024).

Here, we characterise the in vitro substrate specificity of recombinant human HOIL-1, showing that HOIL-1 can efficiently ubiquitinate Ser residues and different saccharides, with weak activity for Thr and no activity for Lys residues. We identify HOIL-1 His510 as a key active site residue that discriminates between ubiquitination of hydroxyl groups in Ser/Thr residues and $\varepsilon$-amine groups in Lys residues. Our work shows that HOIL-1 can ubiquitinate a broad range of saccharides tested, with only small differences in the relative activity. We use human HOIL-1 and an engineered, constitutively active HOIL-1 variant to efficiently generate preparative, milligram amounts of various highly pure ubiquitinated saccharides. These ubiquitinated saccharides will be essential as novel tool compounds and standards to investigate non-proteinaceous ubiquitination by HOIL-1 and other E3 ligases and to test recognition of these ubiquitin signals by specific receptors and their removal by deubiquitinases (DUBs). To this end, in a proof-of-concept experiment, we show that several human DUBs can cleave ubiquitinated maltose in vitro. Together, our work provides new insights into HOIL-1 in vitro activity and specificity and an avenue to develop novel tools to investigate non-proteinaceous ubiquitination.

# Results

## HOIL-1 forms O-linked ubiquitin on Ser/Thr-containing peptides

To compare HOIL-1 E3 ligase activity for different amino acid acceptors, we reconstituted HOIL-1 E3 ligase activity in vitro using the recombinant E1, E2 UbcH7, and full-length human HOIL-1 in the presence of M1-linked di-ubiquitin, previously shown to allosterically activate HOIL-1 (Kelsall et al, 2022; Wu et al, 2022; Wang et al, 2023; Xu et al, 2023). We compare HOIL-1 activity against different pentameric peptides with the general sequence Ac-EGxGN-NH$_2$, where the central amino acid x is either a Lys (K), Ser (S), Thr (T), or Arg (R) residue (Pao et al, 2018). Lys contains an amine as an acceptor, Ser and Thr contain a hydroxyl group acceptor, whereas the

Arg sidechain (bearing a guanidine functional group in the sidechain) is not expected to be an acceptor for ubiquitination.

Our experiments show that HOIL-1 can efficiently attach ubiquitin to the Ser-containing peptide as indicated by disappearance of the ubiquitin-substrate band and appearance of a slower migrating ubiquitin-peptide band via SDS–PAGE (Fig 1A). Under our reaction conditions, half of the ubiquitin is conjugated to the Ser-containing peptide between the 30- and 60-min time points (Fig 1A). We also observe formation of a ubiquitinated peptide with the Thr-containing peptide, albeit to a lower degree with only about 20% of ubiquitin conjugated to the peptide (Fig 1A). In contrast to the activity with the hydroxyl group–containing Ser and Thr peptides, we do not observe ubiquitination of the Lys- and Arg-containing peptides with WT HOIL-1, even though HOIL-1 in this assay is clearly active, as indicated by HOIL-1 autoubiquitination (Fig 1B).

## HOIL-1 His510 enables O-linked ubiquitination and prohibits ubiquitin discharge onto Lys residues

HOIL-1's C-terminal residue, His510, forms part of HOIL-1's catalytic triad and is critical for maltose ubiquitination (Wu et al, 2022; Wang et al, 2023; Xu et al, 2023). RBR E3 ubiquitin ligases like HOIL-1 catalyse ubiquitination in a two-step mechanism where ubiquitin is first transferred from the E2 active site Cys to the E3 active site Cys before being transferred to the substrate (Wenzel et al, 2011; Cotton & Lechtenberg, 2020). We previously showed that mutation of His510 to Ala (H510A) in a HOIL-1 helix-RBR construct does not affect E2-ubiquitin discharge (Wang et al, 2023), indicating that His510 is critical only for the second catalytic step. Interestingly, although a HOIL-1 H510A mutant was inactive in ubiquitinating maltose, we previously observed enhanced autoubiquitination of this mutant compared with WT HOIL-1 (Wang et al, 2023). These observations suggest that HOIL-1 His510 plays a complex role in ubiquitin transfer from the HOIL-1 active site to different substrates.

To further investigate the altered activity of the HOIL-1 H510A mutant and compare the role of His510 in ubiquitinating different acceptors, we compared HOIL-1 WT and H510A activity in the peptide ubiquitination assays. To enable a better comparison with our previous data (Wang et al, 2023), we performed these experiments with the helix-RBR HOIL-1 construct rather than full-length HOIL-1. As before with full-length WT HOIL-1 (Fig 1A), WT helix-RBR HOIL-1 efficiently ubiquitinates the Ser peptide and, to a lesser extent, the Thr peptide (Fig 1C). The slower migrating band disappears upon treatment with hydroxylamine (NH$_2$OH), which breaks oxyester bonds while leaving (iso)peptide bonds intact (Pao et al, 2018), supporting that the peptides are ubiquitinated on the Ser/Thr sidechains. In contrast to WT HOIL-1, the HOIL-1 H510A mutant does not modify the Ser/Thr peptides even when using a higher substrate concentration (Fig 1D). However, we observe strong autoubiquitination with HOIL-1 H510A compared with WT HOIL-1 (Fig 1C and D). Surprisingly, although the HOIL-1 WT autoubiquitination disappears after treatment with hydroxylamine (Fig 1C), most of the HOIL-1 H510A autoubiquitination is resistant to hydroxylamine treatment when assayed on SDS–PAGE under reducing conditions, although we still observe a small amount of hydroxylamine-sensitive autoubiquitination (Fig 1D). Our experiment suggests that most of the autoubiquitination is via isopeptide-linked ubiquitination on a Lys sidechain, rather than Ser/Thr ubiquitination or ubiquitin

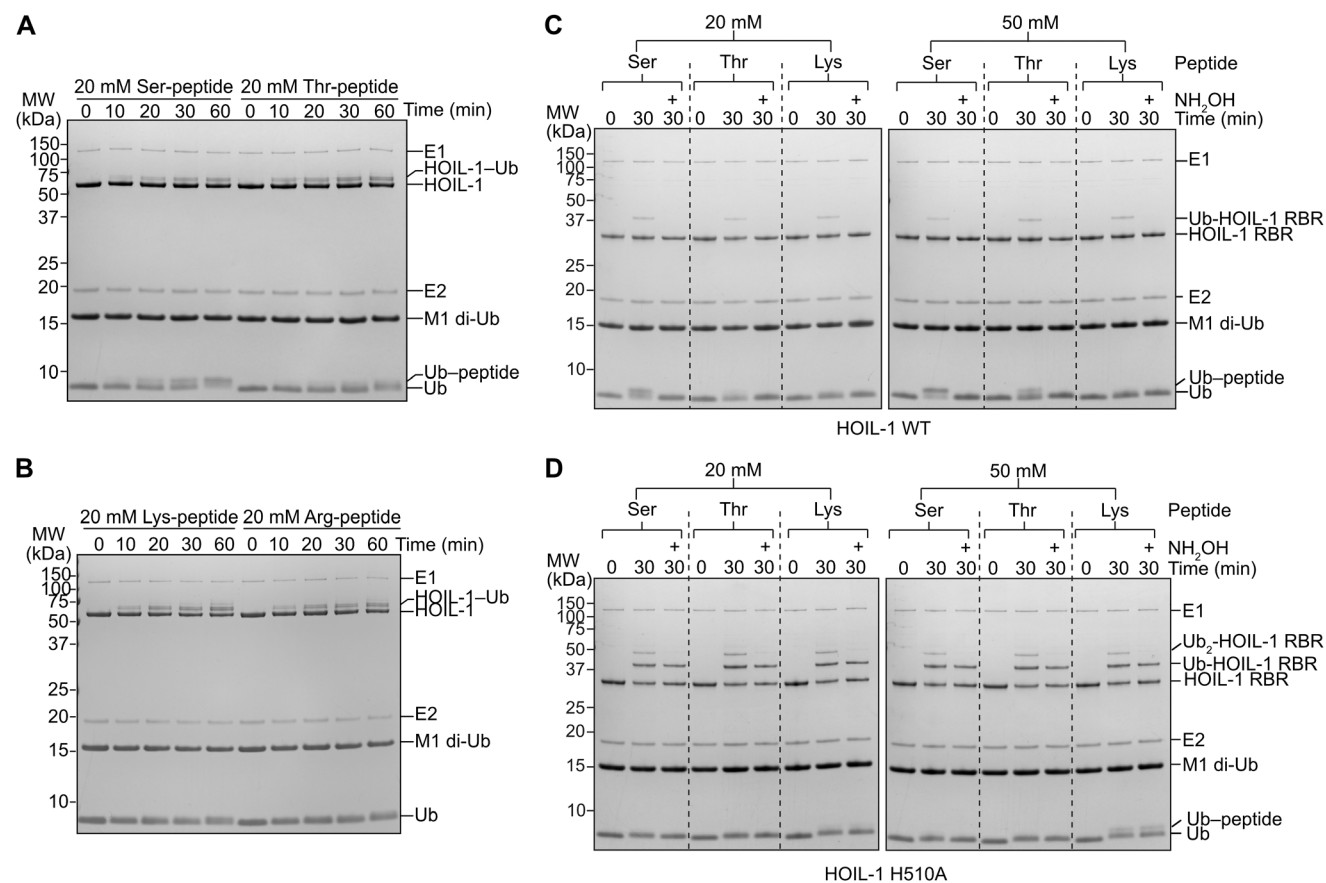

**Figure 1. HOIL-1 ubiquitination of Ser and Thr residues in vitro is dependent on His510.**
**(A)** Peptide ubiquitination assay with 20 mM synthetic Ser- or Thr-containing peptides (Ac-EGxGN-NH$_2$, where x is either Ser or Thr). **(B)** Peptide ubiquitination assays as in panel A with Lys- or Arg-containing peptides. Experiments in panels A and B were performed three times with consistent results. **(C)** Peptide ubiquitination assay with 20 mM (left) and 50 mM (right) synthetic Ser-, Thr-, or Lys-containing peptides and WT HOIL-1. Samples were taken at the indicated time points and analysed via SDS–PAGE under reducing conditions. Half of the samples taken at 30 min were treated with hydroxylamine (NH$_2$OH) as indicated. **(D)** Peptide ubiquitination assay as in panel (C) but with the HOIL-1 H510A mutant. Experiments in panels (C, D) were performed at least twice with consistent results. Representative gels are shown. Ub: ubiquitin. The dotted lines in panels (C, D) are added for readability and do not indicate gel splicing.
Source data are available for this figure.

loaded to the catalytic Cys. We therefore investigated the activity of HOIL-1 WT and H510A with the Lys peptide. As before, we do not observe Lys peptide ubiquitination by WT HOIL-1 (Fig 1C). However, using HOIL-1 H510A, we detect a faint modification of the Lys peptide with 20 mM substrate, with a more pronounced product formation with 50 mM substrate (Fig 1D). These modifications are hydroxylamine-resistant, indicating that they are isopeptide-linked via the lysine sidechain, in line with our observations of HOIL-1 autoubiquitination.

Together, these results confirm that HOIL-1 His510 is required for oxyester-linked ubiquitination, but at the same time prevents ubiquitination of Lys sidechains. His510 therefore is a critical element to fine-tune HOIL-1 activity towards oxyester-linked ubiquitination.

## Ubiquitin binding modulates the HOIL-1 active site conformation

To investigate the importance of His510 for HOIL-1 catalytic activity, we determined the structure of the HOIL-1 RING2 catalytic domain with the active site C460A mutation in the presence of ubiquitin

(Figs 2 and S1, Table 1). The asymmetric unit of our structure contains two copies of the HOIL-1 RING2 domain, one of which is loaded with ubiquitin (Figs 2A and S1A). The overall fold of the HOIL-1 RING2 domain is the same as in previously determined HOIL-1 RING2 structures, with an α-helix preceding the RING2 domain and the HOIL-1–exclusive 2Zn/6Cys binuclear cluster forming the C-terminal portion of the RING2 (Fig S1B) (Wu et al, 2022; Wang et al, 2023; Xu et al, 2023). The ubiquitin Ile44 patch interacts with the HOIL-1 helix, with the ubiquitin C terminus pointing into the HOIL-1 active site (Fig 2B). The ubiquitin therefore mimics the donor ubiquitin as observed in the E2-ubiquitin/HOIL-1 transthiolation complex (Fig S1B) (Wang et al, 2023).

Comparison of the ubiquitin-bound HOIL-1 RING2 with the unbound RING2 in the asymmetric unit enables us to investigate the conformational changes induced by ubiquitin binding. Ubiquitin binding does not globally alter the HOIL-1 RING2 fold, but we observe two distinct changes. First, ubiquitin binding alters the angle between the RING2 domain and its N-terminal helix by 30–40° (Figs 2C and S1C). This observation supports the notion that RBR ligases

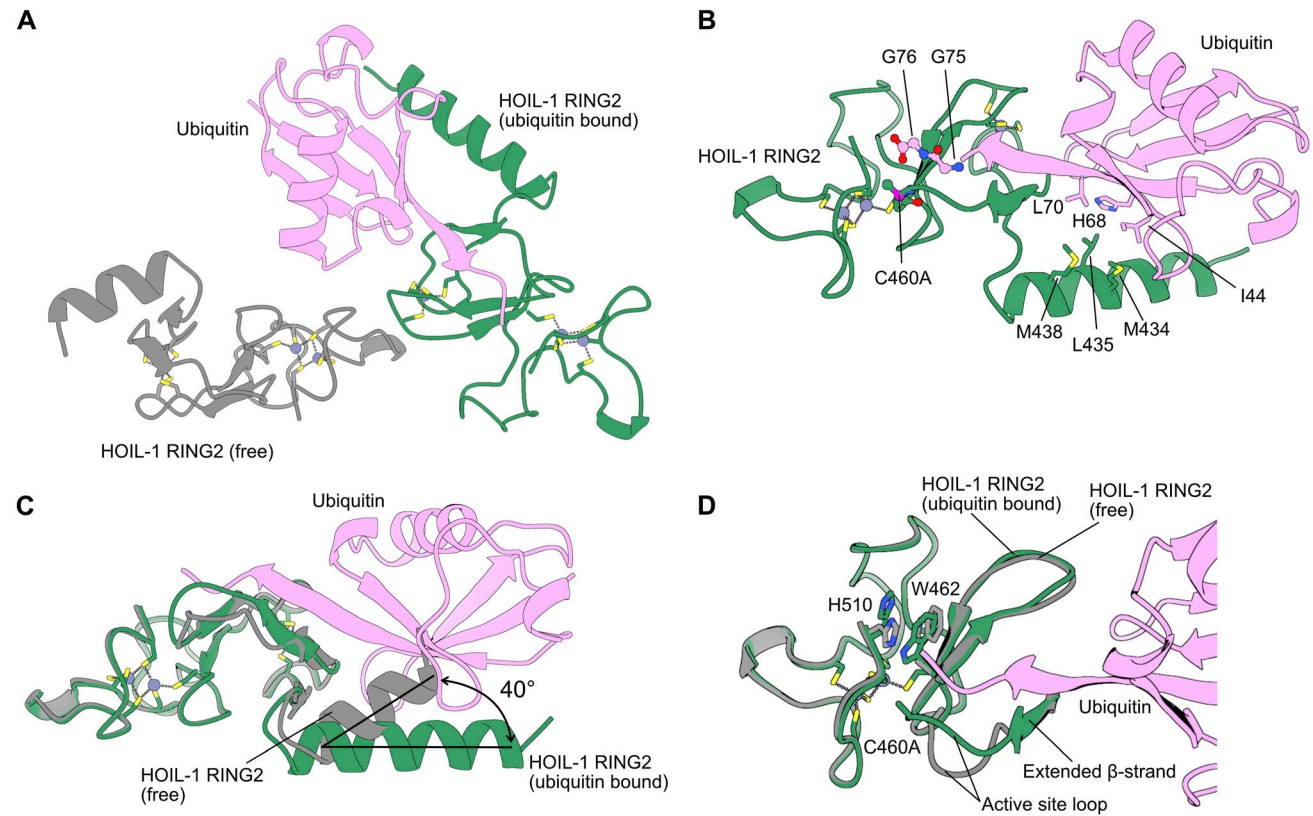

**Figure 2. Crystal structure of the HOIL-1 RING2 domain bound by ubiquitin.**
**(A)** Overview of the crystallographic asymmetric unit shows two HOIL-1 RING2 domains (green, grey) with one of them (green) bound to ubiquitin (pink). Bound zinc ions are shown as purple spheres with zinc ion–coordinating amino acid sidechains shown as sticks. **(B)** Detailed view of the interactions between ubiquitin (pink) and the HOIL-1 RING2 domain (green). Key interacting residues are shown as sticks and labelled. **(C)** Structural superposition of the HOIL-1 RING2 domains in the free state (grey) and bound to ubiquitin (green with ubiquitin in pink) highlights the conformational change between the core RING2 domain and the N-terminal helix upon ubiquitin binding. **(D)** Structural superposition of the HOIL-RING2 domains in the free state (grey) and bound to ubiquitin (green with ubiquitin in pink) highlights the conformational changes in the active site loop, W462 and H510. The active site Cys460 in the structure is mutated to Ala (C460A). The N-terminal part of the HOIL-1 RING2 domain is omitted for clarity.

are highly dynamic enzymes that undergo vast conformational changes throughout their catalytic cycle to allow for binding of their different interaction partners (Walden & Rittinger, 2018; Cotton & Lechtenberg, 2020; Wang et al, 2023). Notably, the conformational changes are not restricted to the N-terminal helix, but ubiquitin binding also rearranges the HOIL-1 active site. The biggest differences are observed in the protein backbone around the active site Cys460 residue (mutated to Ala in the structure), in the active site loop between Gln455 and Asp461, and in the sidechain orientations of Trp462 and His510 (Fig 2D). In the ubiquitin-bound structure, Gln455-Lys457 form an extended β-sheet with the ubiquitin C terminus, whereas the same region in the unbound HOIL-1 RING2 adopts a less structured conformation (Fig 2D). The ubiquitin C terminus also leads to a different rotamer of the Trp462 sidechain, where Trp462 is flipped towards the active site Cys460, leading to rearrangement of the His510 sidechain away from the active site (Fig 2D). These rearrangements in the active site upon ubiquitin binding closely match the conformation previously observed in the HOIL-1/E2-ubiquitin thioester transfer complex structure (Fig S1D) (Wang et al, 2023), suggesting that the observed ubiquitin-bound conformation presents the catalytically active RING2 conformation.

The conformation of the unbound HOIL-1 RING2 closely matches the RING2 of two recently published HOIL-1 structures without ubiquitin (Fig S1E) (Wu et al, 2022; Xu et al, 2023).

## HOIL-1 prefers maltose over Ser/Thr-containing peptides as substrates

As outlined before, HOIL-1 has been reported to ubiquitinate hydroxyl groups in Ser/Thr residues and other biomolecules, such as glucosaccharides. Therefore, after establishing that HOIL-1 most efficiently ubiquitinates hydroxyl groups in Ser residues over Thr residues and does not ubiquitinate Lys residues, we investigated whether HOIL-1 prefers hydroxyl groups in amino acids over hydroxyl groups in oligosaccharides. Our data show that HOIL-1 can ubiquitinate maltose more efficiently than the Ser- or Thr-containing peptides at the same substrate concentrations (Figs 3A and S2A). Notably, even though maltose is a disaccharide consisting of two α-1,4-linked glucose molecules (Fig 3B), we do not observe any evidence of dual ubiquitinated maltose species in our gel-based assays or matrix-assisted laser desorption/ionisation time-of-flight (MALDI-TOF) mass spectrometry (Fig S2A and B) in line

**Table 1.  Crystallographic data collection, processing, and refinement statistics.**

| | HOIL-1 RING2 (C460A)/ubiquitin | HOIL-1 RING2 (C460A)/ubiquitin-maltose |
|---|---|---|
| **Data collection** | | |
| Space group | I $4_1$ 2 2 | I $4_1$ 2 2 |
| Cell dimensions | | |
| a, b, c (Å) | 95.92, 95.92, 187.43 | 96.06, 96.06, 187.43 |
| α, β, γ (°) | 90, 90, 90 | 90, 90, 90 |
| Resolution (Å)[a] | 47.96–2.00 (2.05–2.00) | 46.86–1.78 (1.82–1.78) |
| $R_{merge}$ | 0.214 (2.563) | 0.106 (1.395) |
| I/σI | 8.6 (0.8) | 12.9 (1.3) |
| Completeness (%) | 99.7 (96.6) | 99.9 (98.7) |
| Redundancy | 13.3 (12.0) | 13.5 (12.9) |
| **Refinement** | | |
| Resolution (Å) | 47.96–2.00 | 46.86–1.78 |
| No. of reflections | 29880 | 42215 |
| $R_{work}$/$R_{free}$ | 0.1785/0.2080 | 0.1649/0.1821 |
| No. of atoms (non-H) | | |
| Protein | 1864 | 1904 |
| $Zn^{2+}$ ions | 6 | 6 |
| Other ligands | 16 | 16 |
| Water | 103 | 190 |
| Average B-factors | | |
| Protein | 55.60 | 46.74 |
| Ligand/ion | 69.02 | 62.29 |
| Water | 40.73 | 40.32 |
| RMS deviations | | |
| Bond lengths (Å) | 0.012 | 0.014 |
| Bond angles (°) | 1.063 | 1.211 |
| Ramachandran plot | | |
| Favoured (%) | 97.47 | 97.88 |
| Outliers | 0 | 0 |

[a]Values in parentheses are for the highest resolution shell.

with previous work showing that the larger oligosaccharides maltoheptaose and α-cyclodextrin are mono-ubiquitinated on the C6 hydroxyl group of only one of their glucose subunits (Kelsall et al, 2022; Xu et al, 2023).

### HOIL-1 ubiquitinates different model disaccharides with little specificity

There is a great deal of diversity in oligosaccharide structure because of their complex biochemistry that allows for different linkages of two D-glucose molecules to form 10 disaccharide isomers, including α- and β-linkages, as in maltose (α-1,4) and cellobiose (β-1,4), and linkages between different positions of each sugar unit (regioselectivity): α-1,1 in trehalose, α-1,2 in kojibiose, α-1,3 in nigerose, α-1,4 in maltose, and α-1,6 in isomaltose (Fig 3B).

Although many of these disaccharides are not produced in humans, they enable systematic interrogation of HOIL-1 substrate specificity, or lack thereof. A comparison of maltose (α-1,4) and cellobiose (β-1,4) shows that HOIL-1 can ubiquitinate both disaccharides to a similar level after 30 min (Figs 3C and S2C). Maltose shows a slightly higher initial rate of ubiquitination, although the difference in the initial rate of transfer (up to 10 min) is only about 1.4-fold (Fig 3C).

Next, we compared different disaccharides with regioselectivity. We restricted our analysis to α-glycosidic linkages as found in maltose. As before, we only observe small differences in HOIL-1 activity using maltose or any of the other disaccharides as substrates (Figs 3D and S2C). The biggest difference, 1.4-fold, was observed between maltose (α-1,4) and isomaltose (α-1,6) (Fig 3D). In summary, HOIL-1 can ubiquitinate all disaccharides tested with only marginal differences in the rate of ubiquitination, suggesting

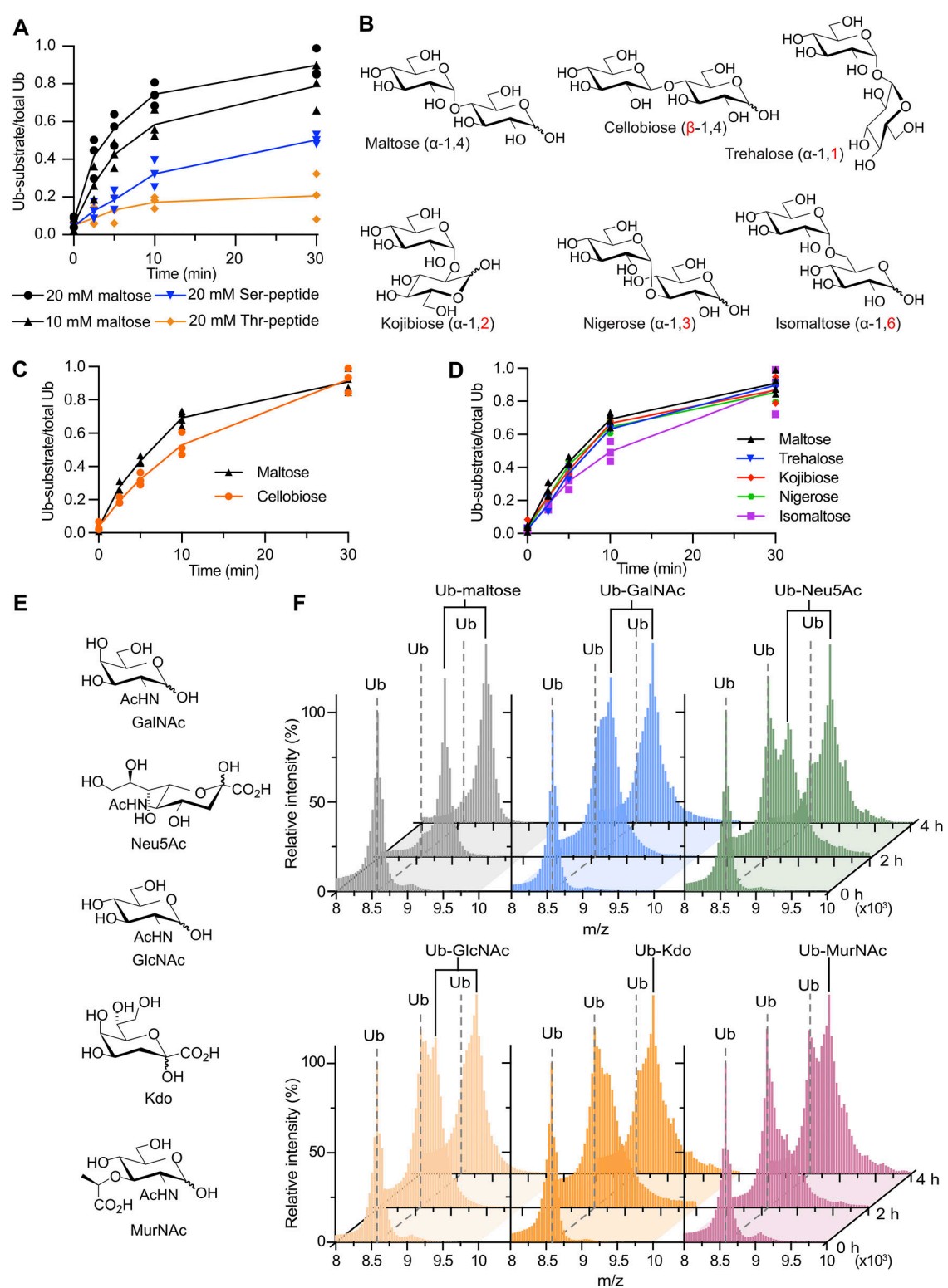

**Figure 3. HOIL-1 ubiquitinates multiple disaccharides and physiological monosaccharides in vitro.**
**(A)** Quantification of HOIL-1–mediated in vitro substrate ubiquitination comparing Ser and Thr peptides with maltose. The proportion of ubiquitinated substrate over total ubiquitin as quantified from Coomassie-stained SDS–PAGE gels is plotted over time. Individual data points and connecting lines of the mean are shown (n = 3).
**(B)** Chemical structures of disaccharide substrates tested. Differences to maltose are highlighted in red font. **(C)** Quantification of HOIL-1–mediated in vitro substrate ubiquitination reactions comparing the disaccharides maltose and cellobiose. Individual data points and connecting lines of the mean are shown (n = 3–4).
**(D)** Quantification of HOIL-1–mediated in vitro substrate ubiquitination reactions comparing disaccharides as indicated. Individual data points and connecting lines of the mean are shown (n = 2–4). Maltose data in panels (C, D) are from the same experiment. **(E)** Chemical structures of the physiological monosaccharide substrates tested.

that HOIL-1 has broad substrate specificity for different model disaccharides.

## HOIL-1 ubiquitinates a variety of physiological monosaccharides in vitro

So far, we systematically investigated HOIL-1 activity on different model disaccharides including many with limited physiological relevance in humans. To investigate more physiologically relevant potential HOIL-1 substrates, we selected multiple saccharides that may be present in a cellular environment in humans (Fig 3E). These include N-acetyl-D-galactosamine (GalNAc) and N-acetyl-D-glucosamine (GlcNAc), which are common sugars found in O-linked protein glycosylation in humans; ketodeoxyoctonic acid (Kdo) and N-acetylmuramic acid (MurNAc) found on the surface of bacterial pathogens that may infect human cells; and N-acetylneuraminic acid (Neu5Ac), a sialic acid that is a common terminal motif on mammalian glycoconjugates. We used the same in vitro assay as before, but instead of trying to resolve and detect products by SDS–PAGE, we used MALDI-TOF MS to distinguish between modified and unmodified ubiquitin species. To simplify analysis via MALDI-TOF, we omitted the allosteric activator M1-linked di-ubiquitin and instead increased the reaction time from 0.5 to 4 h.

Our control experiment with maltose confirms that HOIL-1 can ubiquitinate maltose under these conditions and shows that MALDI-TOF MS can efficiently discriminate between free ubiquitin and ubiquitinated maltose (Fig 3F). Likewise, HOIL-1 ubiquitinates GalNAc, Neu5Ac, GlcNAc, Kdo, and MurNAc within 4 h (Fig 3F), albeit with different efficiency. Although maltose is fully modified, only about 50% of the ubiquitin is conjugated to MurNAc, suggesting that MurNAc ubiquitination by HOIL-1 is less efficient than maltose ubiquitination. The 2-h time point reveals more subtle differences between the different substrates with the near-complete conversion of ubiquitin to ubiquitin-saccharide with maltose and GalNAc (Fig 3F). We only observe ~50% conversion with Neu5Ac, GlcNAc, and Kdo, whereas less than half of the ubiquitin is conjugated to MurNAc (Fig 3F). Together, these experiments show that HOIL-1 can efficiently ubiquitinate a variety of mono- and disaccharides in vitro.

## HOIL-1–mediated production of preparative amounts of ubiquitinated saccharides

Ubiquitination of non-proteinaceous substrates, such as LPS, lipids, and saccharides, is of growing interest in the ubiquitin field (Otten et al, 2021; Kelsall, 2022; Kelsall et al, 2022; Sakamaki et al, 2022; Squair & Virdee, 2022; Zhu et al, 2022, 2024; Dikic & Schulman, 2023; Sakamaki & Mizushima, 2023; Xu et al, 2023; Dearlove et al, 2024). However, tools to study these novel modifications are currently very limited (Lechtenberg & Komander, 2024). Therefore, the scaled-up chemoenzymatic preparation of ubiquitinated non-proteinaceous substrates could provide standards to benchmark new quantitative

methods or probes to identify and characterise novel ubiquitin-substrate binding domains or DUBs. As we observe that HOIL-1 can efficiently and specifically ubiquitinate a wide variety of saccharides, we hypothesised that HOIL-1 would be an ideal enzyme to generate preparative quantities of ubiquitinated non-proteinaceous substrates.

To this end, we modified our in vitro HOIL-1 ubiquitination assay to achieve higher yields of ubiquitinated product and reaction completion to enable purification of the ubiquitinated product. As a proof of principle, we used maltose as a substrate, because we had determined it to be a good substrate in vitro. The key modifications were to use a 10-fold higher concentration of ubiquitin (100 µM) and to increase the reaction time to up to 16 h. We also used an N-terminally His-tagged ubiquitin to enable purification of the ubiquitinated maltose from unreacted maltose and other reaction components.

Comparing the reaction rate of HOIL-1 in a small-scale time-course experiment using 10 or 100 µM His-tagged ubiquitin shows that HOIL-1 efficiently modifies 50% of 10 µM ubiquitin within 2 h with full conversion to ubiquitin-maltose after 16 h (Fig 4A). Under the same enzyme concentrations but with 100 µM ubiquitin, we observe that after 2 h, only about 25% of the ubiquitin is conjugated to maltose (Fig 4A). Incubation overnight (~16 h) results in full ubiquitin conversion to the slower migrating band without evidence of secondary reaction products, for example, ubiquitin chains (Fig 4A). Based on these promising results, we developed a protocol to generate and purify milligram quantities of ubiquitin-maltose (Fig 4B–E). We scaled up the reaction to a 5 ml volume and used 100 µM His-tagged ubiquitin (~1 mg/ml) for a total input of 5 mg ubiquitin. Reaction for 16 h resulted in complete conversion to the slower migrating His-tagged ubiquitin-maltose band (Fig 4C). Subsequent purification via Ni-NTA affinity chromatography resulted in enrichment of His-tagged ubiquitin-maltose (Fig 4C). Final purification by size-exclusion chromatography yielded His-tagged ubiquitin-maltose at excellent purity as determined by SDS–PAGE (Fig 4D) and native mass spectrometry (Fig 4E). Using this protocol, we can typically generate 2.5 mg highly pure ubiquitin-maltose from 5 mg ubiquitin input. Our method is applicable to a wide variety of saccharides including GalNAc, GlcNAc, N-acetyllactosamine (LacNAc), and mucin-type O-glycan Tn, T, and sTn antigens α-linked to serine (Fig S3A–C).

## Application of ubiquitin-maltose to investigate non-proteinaceous ubiquitination in vitro

To demonstrate the utility of our ubiquitinated maltose tool biomolecule, we followed two strategies. First, we attempted to crystallise the HOIL-1 (C460A) RING2 domain with His-ubiquitin-maltose to obtain insights into how HOIL-1 may ubiquitinate maltose. We crystallised HOIL-1 RING/ubiquitin-maltose under the same conditions as HOIL-1 RING2/ubiquitin and successfully determined the X-ray structure to 1.7 Å resolution (Table 1). As with the

---

**(F)** MALDI-TOF analysis of HOIL-1–mediated in vitro substrate ubiquitination reactions of the substrates shown in panel (E). Experiments were performed twice with consistent results. Exemplary data are shown. Ub: ubiquitin.
Source data are available for this figure.

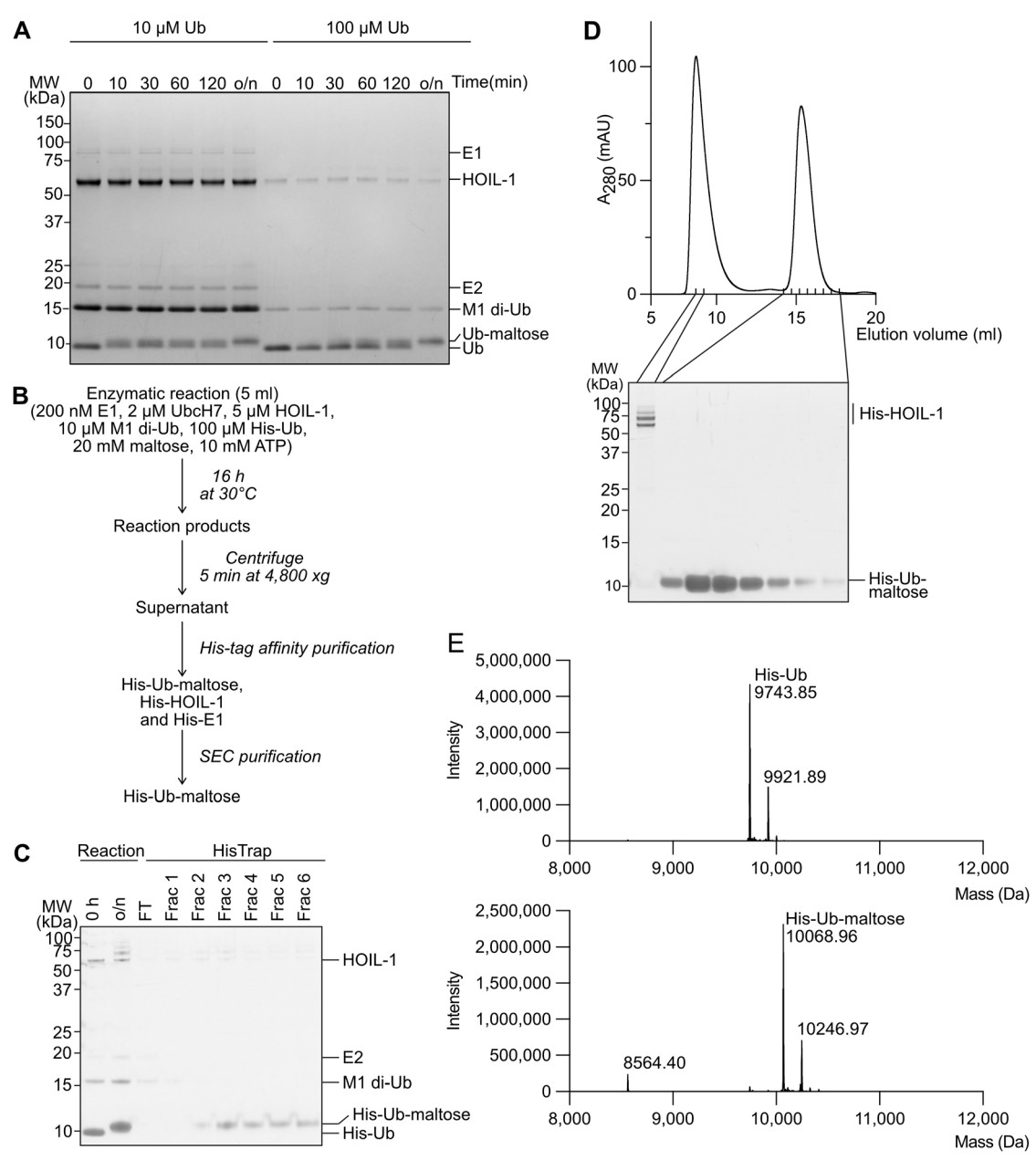

**Figure 4. Large-scale preparation and purification of ubiquitinated maltose.**
**(A)** Time course showing HOIL-1–mediated ubiquitination of maltose using low (10 μM) and higher (100 μM) ubiquitin concentrations. To enable comparison of ubiquitin (Ub) and ubiquitin-maltose bands between the different samples, the samples containing 100 μM were diluted 10-fold before SDS–PAGE analysis. o/n, overnight. **(B)** Workflow for large-scale ubiquitin-maltose preparation and purification. **(C)** Coomassie-stained SDS–PAGE gel showing successful large-scale ubiquitin-maltose generation (Reaction) and His-tag affinity purification step (HisTrap). Frac: fractions. **(D)** Size-exclusion chromatography purification of His-ubiquitin-maltose after HisTrap purification. **(E)** Intact mass spectrometry of His-ubiquitin (top) and purified His-ubiquitin-maltose (bottom). The mass difference of 325 D confirms mono-ubiquitination of a single maltose molecule. The slightly heavier minor species in the two spectra likely represent α-N-6-phosphogluconoylation of the His-tag, often identified in His-tagged proteins expressed in *E. coli* (Geoghegan et al, 1999).
Source data are available for this figure.

HOIL-1 RING2/ubiquitin structure, the crystallographic asymmetric unit contains two HOIL-1 RING2 molecules and one ubiquitin molecule bound in the donor ubiquitin site of one HOIL-1 RING2 domain (Fig S4A and B). Although we can resolve the ubiquitin C terminus, we do not observe clear electron density for the ubiquitin-conjugated maltose molecule and therefore were unable to build the maltose molecule into the model (Fig S4C), despite intact mass spectrometry confirming that the vast majority of the ubiquitin in crystals grown under identical conditions is conjugated to maltose (Fig S4D). This suggests that maltose does not form specific stable interactions with the HOIL-1 RING2 domain, in line with the broad substrate specificity we observe for HOIL-1 in vitro.

**A**

| Reagent | Lys-linked | M1-linked | Thr-linked | Ub-maltose |
|---------|-----------|-----------|-----------|-----------|
| USP2 | all | | | |
| OTUD1 | K63 | | | |
| OTULIN | | | | |
| Cezanne | K11 | | | |
| ATXN3 | | | | |
| NH₂OH | | | | |

**B**

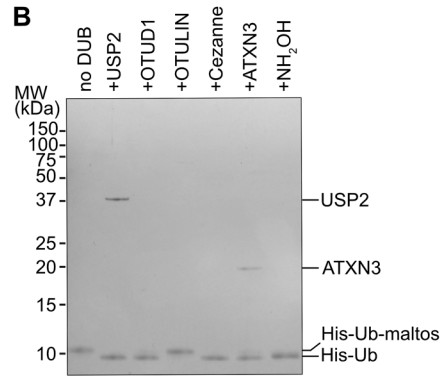

**Figure 5. Proof-of-principle DUB assay towards ubiquitin-maltose hydrolysis.**
**(A)** Table summarising known activities of DUBs and hydroxylamine (NH₂OH) against different amino acid ubiquitin linkages (Lys, M1, Thr) from the literature and results from our analysis with Ub-maltose. Grey cells indicate activity with the indicated species. For Lys chain type–specific DUBs, the cleaved linkages are indicated. **(B)** Ubiquitin-maltose DUB assay. His-ubiquitin-maltose was incubated with different DUBs as indicated and analysed via SDS–PAGE. The first lane contains untreated sample, and the sample in the last lane was treated with hydroxylamine (NH₂OH) to fully cleave oxyester-linked His-ubiquitin-maltose. DUBs were used at different, optimal concentrations (Hospenthal et al, 2015); hence, only DUBs used at high concentrations (USP2, ATXN3) are resolved on the gel. Ub: ubiquitin. Source data are available for this figure.

In a separate approach, we used ubiquitin-maltose to investigate a small panel of DUBs to identify which DUBs can cleave the non-proteinaceous substrate ubiquitin-maltose. Our array consists of USP2, OTUD1, OTULIN, Cezanne, and ATXN3. USP2 features broad substrate specificity that can cleave all seven Lys-linked ubiquitin chains (K6, K11, K27, K29, K33, K48, and K63), M1-linked ubiquitin, and oxyester-linked Thr-ubiquitin (Hospenthal et al, 2015; De Cesare et al, 2021). The ubiquitin chain type–specific DUBs OTUD1 (K63-specific), OTULIN (M1-specific), and Cezanne (K11-specific) do not cleave ubiquitinated threonine, whereas ATXN3 does not cleave ubiquitinated lysine but can cleave ubiquitinated threonine (Hospenthal et al, 2015; De Cesare et al, 2021). As a positive control for oxyester-linked ubiquitin, we used hydroxylamine treatment. We first verified DUB activity against a panel of amine-conjugated ubiquitin chains (K63-, K11-, and M1-linked) and observed the expected activities for each DUB (Figs 5A and S5). When we incubated the same DUBs with the purified His-ubiquitin-maltose substrate, we observed that the non-specific USP2 can cleave ubiquitin-maltose, as can the positive control hydroxylamine, as expected (Fig 5A and B). Out of the chain linkage–specific DUBs, OTULIN is unable to cleave ubiquitin-maltose as expected (Fig 5A and B). Surprisingly, OTUD1 and Cezanne, which are not active against the Ub-threonine oxyester (De Cesare et al, 2021), can cleave the ubiquitin-maltose oxyester (Fig 5A and B). Finally, ATXN3 also can cleave the ubiquitin-maltose oxyester linkage (Fig 5A and B). These proof-of-principle experiments exemplify that our HOIL-1–catalysed ubiquitin-maltose can be used for enzymatic DUB assays in vitro. The observation that not all DUBs tested in our assay hydrolyse ubiquitin-maltose highlights that ubiquitin-maltose cleavage is a specific reaction. In line with this, OTUD1 and Cezanne, with reported poor activity against the oxyester-linked substrate ubiquitin-Thr (De Cesare et al, 2021), cleave ubiquitin-maltose in our assay. Our observations form the basis to investigate DUB activity and specificity for non-proteinaceous substrates more thoroughly.

### Generation of a constitutively active HOIL-1 variant enables efficient maltose ubiquitination in the absence of allosteric M1-linked di-ubiquitin

HOIL-1 is fully active only in the presence of allosteric M1-linked di-ubiquitin (Kelsall et al, 2022; Wang et al, 2023; Xu et al, 2023),

necessitating the addition of high concentrations of di-ubiquitin in our assay, or increasing the reaction times. This complicates our in vitro ubiquitination workflow, may introduce artefacts, or may require additional purification steps to remove di-ubiquitin. Therefore, we aimed to generate a constitutively active HOIL-1 by fusing M1-linked di-ubiquitin to the HOIL-1 N terminus (M1-di-Ub-HOIL-1, Fig 6A). A similar approach has previously been implemented in a cellular context for the RBR E3 ligase HOIP, which is also activated by M1 di-ubiquitin (Wegmann et al, 2022). We initially used the native human M1 di-ubiquitin protein sequence with different linker lengths between di-ubiquitin and HOIL-1. Although we were able to recombinantly express these constructs in *Escherichia coli*, we observed large amounts of cleaved M1 di-ubiquitin and free HOIL-1, which made it challenging to obtain a homogeneous fraction of M1-di-Ub-HOIL-1 (Fig S6A). In our final design, we therefore completely removed the linker between M1-di-ubiquitin and HOIL-1 and replaced Gly76 in both ubiquitin moieties with Val (G76V). This final construct expressed and purified almost as well as HOIL-1 (Fig S6A and B). Importantly, when we compared the activity of the M1-di-Ub-HOIL-1 to HOIL-1 in the presence and absence of additional M1-linked di-ubiquitin, we observed that M1-di-Ub-HOIL-1 without exogenous M1-linked di-ubiquitin is as active as HOIL-1 in the presence of the allosteric activator and is not further activated by the addition of exogenous M1-linked di-ubiquitin (Fig 6B). This experiment shows that M1-di-Ub-HOIL-1 is a constitutively active enzyme that can replace HOIL-1 and M1-linked di-ubiquitin to streamline our ubiquitin-maltose generation system.

## Discussion

Ubiquitination of non-proteinaceous substrates including LPSs, sugars, ADPR, ribonucleotides, and lipids is a rapidly emerging theme in the ubiquitin field (Otten et al, 2021; Kelsall et al, 2022; Sakamaki et al, 2022; Zhu et al, 2022, 2024; Dearlove et al, 2024). The RBR E3 ligase HOIL-1 is one of the E3 ubiquitin ligases that has been implicated in ubiquitinating maltose and glycogen, in addition to its previously reported activity in ubiquitinating Ser, Thr, and Lys residues in proteins (Kelsall et al, 2019, 2022; Fuseya et al, 2020; Rodriguez Carvajal et al, 2021; McCrory et al, 2022; Wang et al, 2023; Xu et al, 2023).

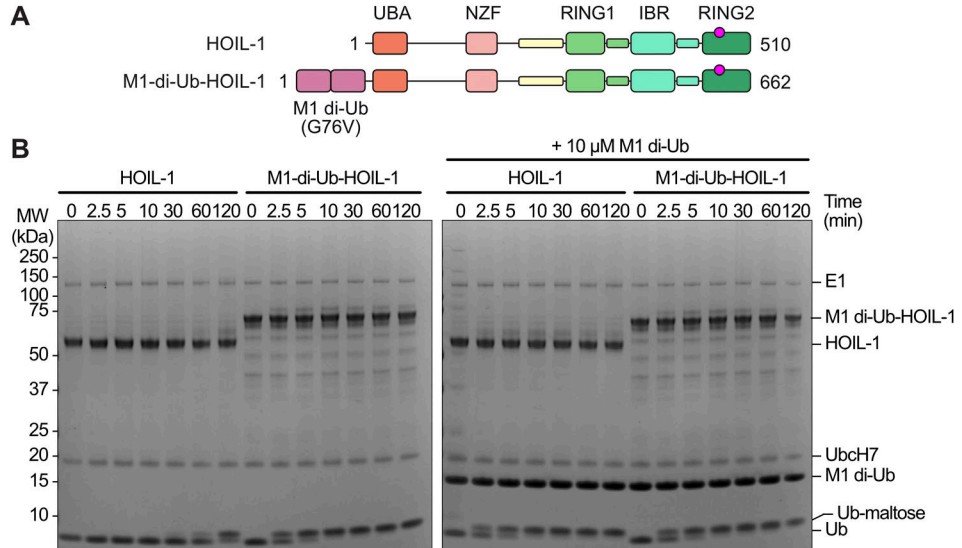

**Figure 6. Design of a constitutively active HOIL-1 variant.**
**(A)** Domain organisation of human HOIL-1 (top) and the constitutively active M1-di-Ub-HOIL-1 fusion protein that contains an N-terminal M1-linked di-ubiquitin in which glycine 76 in both protomers is mutated to valine (G76V).
**(B)** Maltose-ubiquitination assay comparing HOIL-1 with the M1-di-Ub(G76V)-HOIL-1 fusion protein in the presence and absence of additional allosteric M1-linked di-ubiquitin. Ub: ubiquitin.
Source data are available for this figure.

Our work here provides further quantitative insights on the catalytic mechanism of the RBR E3 ubiquitin ligase HOIL-1 and its relative substrate specificity. Using in vitro ubiquitination assays on model peptide substrates and an array of saccharides, we show that HOIL-1 has a strong preference for ubiquitinating hydroxyl groups in Ser sidechains, maltose and other saccharides with weaker activity for ubiquitinating Thr and no detectable activity for the Lys-containing model peptide.

Our biochemical analysis identifies the catalytic His510 residue in HOIL-1's C terminus as a critical factor for Ser ubiquitination. The HOIL-1 H510A mutant is unable to ubiquitinate the Ser model peptide, but, surprisingly, gains activity to ubiquitinate Lys residues in the model peptide and on HOIL-1 itself. Our crystal structure of the HOIL-1 RING2 domain with and without ubiquitin shows that His510 is part of a dynamic HOIL-1 catalytic centre that includes the loop containing the catalytic Cys460 and Trp462. Our structure demonstrates that ubiquitin binding induces a rearrangement of these regions into what is likely the active conformation that is also observed in previous structures of the HOIL-1/E2-ubiquitin thioester transfer complex. Our biochemical and structural analyses, together with our previous observation (Wang et al, 2023) that His510 is dispensable for HOIL-1's first catalytic step (ubiquitin transfer from E2 to E3), but critical for ubiquitin transfer from the E3 to the substrate, suggest that His510 acts as a general base and is required to deprotonate and activate the incoming acceptor nucleophile. Compared with amines, hydroxyl groups are weaker nucleophiles and therefore critically require activation by HOIL-1's His510 for efficient ubiquitin transfer to the substrate. Mutation of His510 to Ala removes this activating effect, thereby preventing ubiquitin transfer from the E3 active site to the substrate hydroxyl group. The E3 ubiquitin thioester then has two options: hydrolysis of the thioester by a solvent molecule or ubiquitin discharge to an amine, either on a Lys sidechain in HOIL-1 itself or on a Lys in the substrate (if present). This results in the apparent specificity switch from Ser to Lys for the H510A mutant compared with WT HOIL-1. Interestingly, the importance of an active site His residue for

hydroxyl group ubiquitination has recently been described for non-canonical E2 ubiquitin–conjugating enzymes Ube2J2 and Ubc6 (Abdul Rehman et al, 2024; Swarnkar et al, 2024). The fact that we still observe O-linked (hydroxylamine-sensitive) autoubiquitination of HOIL-1 H510A may be due to one of two reasons. The local environment of a surface-exposed HOIL-1 Ser/Thr residue may reduce the sidechain's pkA and thereby lead to an increased fraction of the deprotonated sidechain that can be more efficiently ubiquitinated even in the absence of the catalytic His510. Alternatively, a Ser/Thr residue close to the HOIL-1 active site may be ubiquitinated over time because of its proximity, even if the ubiquitination reaction is inefficient.

Our work further shows that HOIL-1 has broad substrate specificity against different mono- and disaccharides. Previous works show that HOIL-1 ubiquitinates the C6 hydroxymethyl ($CH_2OH$) moiety in a glucose subunit of maltoheptaose (Kelsall et al, 2022). All successful substrates tested in our work contain a hydroxymethyl moiety, either in the Ser sidechain or in the various saccharides. In the saccharides, the $CH_2OH$ moiety is mostly found in the C6 position, with the exceptions of Kdo, where the hydroxymethyl moiety includes C8, and Neu5AC, where it includes C9 (Fig 3B and E). The major caveat of the in vitro experiments here and elsewhere (Kelsall et al, 2022; Wang et al, 2023; Xu et al, 2023) is that very high substrate concentrations of 20 mM are used, which may have limited physiological relevance. As we have argued before, just because an E3 ligase can ubiquitinate a substrate in vitro does not mean that it does so in a cellular environment (Lechtenberg & Komander, 2024). On the flip side, high local concentrations of maltose subunits, as found in the proposed HOIL-1 substrate glycogen, may provide the right environment for efficient HOIL-1–mediated ubiquitination of these biomolecules in the cell. We also observe HOIL-1–mediated in vitro ubiquitination of saccharides found in high concentration on the surface of pathogens, for example, Kdo, a component of bacterial LPS, and MurNAc, a component of bacterial peptidoglycan. This observation may hint at additional functions of HOIL-1 in ubiquitinating intracellular

pathogens, as previously described for the E3 ligase RNF213 (Otten et al, 2021), and in line with LUBAC's role in innate immune signalling. In all cases, to overcome the limitations of these in vitro experiments, it will be critical to develop novel technologies that can detect oxyester-linked non-proteinaceous substrates, such as ubiquitinated sugars in the cell.

Recent publications report that current mass spectrometry methods may not be able to detect oxyester-conjugated ubiquitin species because of their reduced stability compared with isopeptide-linked ubiquitinated species (Yoshida et al, 2024). Our approach to generate ubiquitinated model products will provide benchmarks that can be used to develop and optimise mass spectrometry workflows that are compatible with ubiquitinated sugars, for example, by spiking-in our model products in cell lysates and detecting the expected masses. In addition, ubiquitinated sugars generated with our protocol can be used to investigate DUB activity and specificity as shown in our proof-of-principle work. Our experiments show that ubiquitin-maltose cleavage is a specific enzyme-catalysed reaction, as not all DUBs tested cleave the substrate. However, surprisingly, DUBs that previously were shown to be inactive against ubiquitin-Thr oxyester bonds (OTUD1 and Cezanne [De Cesare et al, 2021]) can cleave the ubiquitin-maltose oxyester. This highlights the specificity of the oxyester hydrolysis and substrate specificity of these DUBs and the importance of comparing various oxyester-containing species as we have performed here for HOIL-1. To further delineate the differences in DUB activities against various oxyester-linked ubiquitin species, more sophisticated experiments with different enzyme concentrations and time courses are required in the future. Our proof-of-principle study provides a strong foundation for these experiments. In addition to the assays presented here, we envision that our HOIL-1–generated ubiquitin-saccharide species will be valuable for investigating other aspects of non-proteinaceous ubiquitination, for example, to pull down specific ubiquitin-sugar binding proteins, which could then be identified by proteomics methods, supplementing other emerging technologies specific for oxyester-linked ubiquitin (Szczesna et al, 2024).

In summary, our work provides novel insights on the catalytic activity of the RBR E3 ubiquitin ligase HOIL-1 and its non-canonical ubiquitination activity towards non-proteinaceous substrates. We highlight how HOIL-1 and an engineered constitutively active HOIL-1 variant can be used to generate tool compounds and standards to interrogate non-proteinaceous ubiquitination and to develop urgently needed techniques in this burgeoning field.

# Materials and Methods

## Plasmids and reagents

The plasmid for the bacterial expression of full-length human His-3C-HOIL-1 (pOPINB-HOIL-1 full-length) is available from Addgene (Plasmid #193858) (Wang et al, 2023). His-3C-HOIL-1 RING2 domain (residues 425–510) in pOPINB was cloned from this plasmid and the C460A mutation introduced via the Q5 site-directed mutagenesis kit (NEB). UBE1, UbcH7, native ubiquitin, non-cleavable His-ubiquitin, and M1 di-ubiquitin constructs were described previously

(Lechtenberg et al, 2016; Cotton et al, 2022; Wang et al, 2023). The M1-di-Ub(G76)-HOIL-1 fusion construct was cloned via in-fusion cloning by introducing the sequence of M1-di-Ub(G76V) into the pOPINB-HOIL-1 full-length plasmid. The pOPINB-M1-di-Ub(G76V)-HOIL-1 fusion plasmid for bacterial expression is available from Addgene (Plasmid # 229539). Saccharides were purchased from Biosynth Carbosynth. Tn, sTn, and T glycosyl amino acids were synthesised according to a method described by Corcilius & Payne (2013).

## Protein expression and purification

Full-length HOIL-1 was expressed in *E. coli* Rosetta(DE3)pLacI, and all other recombinant proteins including the M1-di-Ub(G76V)-HOIL-1 fusion were expressed in *E. coli* BL21(DE3). Bacterial cultures were grown in 2x YT media with antibiotics until $OD_{600}$ ~0.8. Protein expression was induced with 0.5 mM IPTG (Golden Biotechnology). Cultures expressing the HOIL-1 protein were in addition supplemented with 0.5 mM $ZnCl_2$ at the time of induction. The media for the M1-di-Ub(G76V)-HOIL-1 fusion expression were in addition supplemented with 15 ml of 750 mM mannitol per litre of media to reduce lower molecular weight contaminants found during purification. Cultures were grown overnight at 20°C before harvesting by centrifugation.

For purification, cells were lysed with the addition of lysozyme and sonication in the presence of protease inhibitors (PMSF and leupeptin), DNase I, and $MgCl_2$. Cell lysates were cleared by centrifugation. To purify His-tagged E2, ubiquitin, M1 di-ubiquitin, and full-length and RING2 HOIL-1 proteins, cleared cell lysates were applied to Ni-NTA His-bind resin (Merck Millipore) equilibrated in gravity columns. His-tagged proteins were eluted with high-salt purification buffer (50 mM Tris, pH 8, 500 mM NaCl, 10% sucrose, and 10% glucose) supplemented with 300 mM imidazole. Eluted proteins were dialysed overnight in the presence of 1/100 (vol/vol) His-HRV 3C protease to remove the His-tag. Cleaved proteins were applied to Ni-NTA His-bind resin to remove His-HRV 3C protease before purification by size-exclusion chromatography (Superdex 75 16/600 pg or Superdex 200 16/600 pg; Cytiva) using the AKTA pure chromatography system (Cytiva).

The His-3C-M1-di-Ub(G76V)-HOIL-1 fusion protein was purified basically as described above for full-length HOIL-1 but with an additional wash step with 30 mM imidazole in high-salt purification buffer before elution in high-salt purification buffer supplemented with 300 mM imidazole, and the His-tag was not cleaved.

K11 tri-ubiquitin and K63 tetra-ubiquitin were generated enzymatically from mono-ubiquitin with ubiquitin linkage–specific enzymes as described previously (Michel et al, 2018). The DUBs USP2, OTUD1, OTULIN, Cezanne, and ATXN3 were expressed in *E. coli* and purified as previously described (Hospenthal et al, 2015). All purified proteins were concentrated and snap-frozen in liquid nitrogen before storing at −80°C.

## Peptide ubiquitination assays

A reaction master mix was made with 200 nM E1, 2 $\mu$M UbcH7, 5 $\mu$M HOIL-1, 10 $\mu$M blocked M1 di-ubiquitin, and 10 $\mu$M native ubiquitin in

DPBS buffer (Gibco) containing 10 mM MgCl$_2$ and 0.5 mM TCEP. 200 mM peptide stocks (20 mM final concentration; GenScript Biotech) were mixed with 0.1 M ATP (pH 7, 10 mM final concentration) and added to the master mix to initiate the reaction. Reactions were incubated at 30°C and sampled at indicated time points by mixing with 1x LDS sample buffer (NuPAGE) supplemented with 0.1 M DTT. Reactions in Fig 1C and D were further treated with 1.5 M NH$_2$OH (467804; Sigma-Aldrich) as indicated for 30 min at 30°C before the addition of 1x LDS sample buffer supplemented with 0.1 M DTT. All samples were denatured at RT for 5 min and analysed by SDS–PAGE (1 mm 12% NuPAGE Bis-Tris Mini Protein Gels; Invitrogen) followed by Coomassie staining. Gels were imaged on a ChemiDoc MP Imaging system (Bio-Rad) and band intensities quantified with ImageLab (Bio-Rad). Plots were generated in Prism 10 (GraphPad).

## Saccharide ubiquitination assays

A reaction master mix was made with 200 nM E1, 2 $\mu$M UbcH7, 5 $\mu$M HOIL-1, 10 $\mu$M blocked M1 di-ubiquitin, and 10 $\mu$M native ubiquitin in DPBS buffer (Gibco) containing 10 mM MgCl$_2$ and 0.5 mM TCEP. 200 mM stocks of individual saccharides dissolved in DPBS buffer were added into individual reactions to reach a 20 mM final concentration. 0-min samples were taken by mixing with 1x LDS sample buffer supplemented with 0.1 M DTT. 10 mM ATP (pH 7) was added to initiate the reactions. Reaction samples were incubated at 30°C, whereas indicated time-point samples were taken by mixing with 1x LDS buffer supplemented with 0.1 M DTT. All samples were denatured at RT for 5 min and analysed by SDS–PAGE followed by Coomassie staining.

The assays for the physiological saccharides were performed in a similar manner, but without the addition of the allosteric activator M1 di-ubiquitin and with increased reaction times as indicated. Samples for MALDI-TOF analysis were taken at 0, 2, and 4 h by acidification with 0.4% TFA (Sigma-Aldrich).

## Mass spectrometry

### MALDI-TOF
A previously described protocol was followed to prepare MALDI samples (Signor & Boeri Erba, 2013). In brief, the acidified sample was diluted 1:1 vol/vol with ACN/FA solution (acetonitrile mixed with a 5% vol/vol formic acid in water in a 70:30 vol/vol ratio). The MALDI target was pretreated with a thin layer of saturated $\alpha$-CHCA ($\alpha$-cyano-4-hydroxycinnamic acid) dissolved in ACN/FA solution. 20 mg/ml DHB solution was prepared in ACN/TFA solution (acetonitrile mixed with 0.1% vol/vol trifluoroacetic acid in water in a 70:30 vol/vol ratio). 0.5 $\mu$l diluted sample was deposited onto the target followed by 0.5 $\mu$l $\alpha$-CHCA/DHB solution (1:1 vol/vol) and air-dried. Data were acquired on a Bruker microflex using 100% laser power with no detection gain. Data were automatically analysed by software flexAnalysis 3.4 (Bruker) and replotted using Prism 9 (GraphPad).

### Intact MS
Purified His-ubiquitin and His-ubiquitin-maltose were diluted to 20 $\mu$M using Milli-Q H$_2$O. All samples were centrifuged for 5 min at 13,000$g$ and 4°C to remove aggregates. Proteins were separated by reversed-phase chromatography on a 25-cm ProSwift RP-4H monolith column (Thermo Fisher Scientific) using a micro-flow HPLC (Ultimate 3000). The HPLC was coupled to a maXis II Q-TOF mass spectrometer (Bruker) equipped with an ESI source. Proteins were loaded directly onto the column for online buffer exchange at a constant flow rate of 50 $\mu$l/min with buffer A (99.9% ultrapure water, 0.1% formic acid) using a valve to direct flow to waste. After 10 min, the valve was switched to direct flow to the mass spectrometer, and proteins were eluted with a 10-min linear gradient from 2 to 90% buffer B (99.9% acetonitrile, 0.1% formic acid). The maXis II Q-TOF was operated in full MS mode using Compass Hystar 5.1 with the following settings: mass range 100–3,000 m/z, capillary voltage 4,500 V, dry gas 4 litres/min, dry temperature 200°C. Data were analysed with DataAnalysis (version 5.2; Bruker), and proteins were deconvoluted using the maximum entropy method. Deconvoluted spectra were exported and replotted using Prism 10 (GraphPad).

## DUB assay

Indicated DUBs were prediluted and incubated with DUB dilution buffer containing 25 mM Tris, pH 7.5, 150 mM NaCl, and 10 mM DTT. M1 di-ubiquitin, K11 tri-ubiquitin, and K63 tetra-ubiquitin were mixed with DUB reaction buffer containing 50 mM Tris, pH 7.5, 50 mM NaCl, and 5 mM DTT. Similarly, His-ubiquitin-maltose was mixed with DUB reaction buffer. To set up the control reaction, the polyubiquitin mixture containing 5 $\mu$M of each chain type was split and mixed with individual DUBs. The final concentration for each DUB is USP2 2 $\mu$M, OTUD1 0.2 $\mu$M, OTULIN 0.1 $\mu$M, Cezanne 0.2 $\mu$M, and ATXN3 2 $\mu$M. DUBs were mixed at the validated concentrations with 5 $\mu$M His-ubiquitin-maltose. All reactions were incubated at RT for 30 min. Reactions were stopped by adding 1x LDS sample buffer supplemented with 0.1 M DTT. Samples were resolved by SDS–PAGE followed by Coomassie staining.

## Large-scale ubiquitinated saccharide production and purification

To enzymatically prepare ubiquitin-maltose, a reaction containing 200 nM E1, 2 $\mu$M UbcH7, 5 $\mu$M HOIL-1, 10 $\mu$M blocked M1 di-ubiquitin, 100 $\mu$M His-ubiquitin, and 20 mM maltose was made up in DPBS buffer supplemented with 10 mM MgCl2 and 0.5 mM TCEP in 5 ml final volume. The reaction was initiated by the addition of 10 mM ATP, incubated at 30°C, and sampled as indicated to check on reaction completeness by SDS–PAGE. After overnight (~16 h) incubation, the reaction solution was clarified by centrifugation (5 min at 4,800$g$) and the supernatant was applied to a 1-ml HisTrap HP column (Cytiva) equilibrated in DPBS. The bound resin was washed with DPBS buffer and block-eluted with DPBS buffer containing 300 mM imidazole. The eluates were examined by SDS–PAGE, pooled, and concentrated for further size-exclusion chromatography purification on a Superdex 75 Increase 10/300 column (Cytiva) equilibrated in DPBS. Purified His-ubiquitin-maltose was concentrated, aliquoted, and snap-frozen in liquid nitrogen, then stored at –80°C.

### Crystallisation

Purified HOIL-1 helix-RING2 (residues 425–510 with the C460A mutation) was mixed with His-tagged ubiquitin or His-tagged ubiquitin-maltose in a 1:2 M ratio. The protein mixture was concentrated using an Amicon Ultra 3 kD MW cut-off device (Merck Millipore) to ~15 mg/ml and used to set up commercial crystallisation screens using the sitting drop vapour diffusion method at RT. Initial crystal hits were obtained from the Natrix (Hampton) screen in a condition containing 0.01 M $MgCl_2$, 0.05 M MES, pH 5.6, and 1.8 M $LiSO_4$ by mixing 100 nl protein with 100 nl reservoir solution. Hit crystals were optimised by titrating $MgCl_2$ (0.01–0.04 M) and varying MES, pH from 5.2 to 6.2. The best diffracting crystals grew in a pH range from 5.2 to 6.0 and various $MgCl_2$ concentrations. The HOIL-1 helix-RING2/Ub-maltose structure was determined from a crystal grown in 0.02 M $MgCl_2$, 0.05 M MES, pH 6, and 1.8 M $LiSO_4$. The crystal was dehydrated by replacing the reservoir solution with 2.5 M $LiSO_4$ and incubation overnight. The HOIL-1 helix-RING2/Ub structure was determined from a crystal grown in 0.01 M $MgCl_2$, 0.05 M MES, pH 6.2, and 1.8 M $LiSO_4$. All crystals were rinsed in fresh reservoir solution and directly flash-frozen in liquid nitrogen.

To prepare crystal samples for intact mass spectrometry, crystals were harvested from two different crystallisation experiments about 2 wk after set-up, washed twice in reservoir solution, and dissolved in buffer (0.1 M MES, pH 6, 150 mM NaCl or 10 mM Hepes, pH 7.9, 100 mM NaCl, 1 mM TCEP). About 10 crystals from a single drop each were collected in 10 $\mu$l buffer, and 7 $\mu$l was analysed by intact MS as described above.

### Diffraction data collection, processing, model building, and refinement

Crystal diffraction data were collected at the Australian Synchrotron MX2 beamline (Aragao et al, 2018) at a wavelength of 0.9537 Å and 100 K temperature. Datasets were indexed and integrated with XDS (Kabsch, 2010) and merged with AIMLESS (Evans & Murshudov, 2013) using the automated processing pipeline at the Australian Synchrotron. Phaser (McCoy et al, 2007) was used to solve the structure by molecular replacement. The initial search was performed using two copies of HOIL-1 RING2 (residues 444–510) molecules from the known HOIL-1 structure (PDB ID 8EAZ) (Wang et al, 2023) and ubiquitin (PDB ID 1UBQ) (Vijay-Kumar et al, 1987) each. Two HOIL-1 molecules and one ubiquitin molecule were successfully found. The RING2 N-terminal helix of chain A (residues 425–443) and chain B (residues 431–443) was manually built. Iterative cycles of model building and refinement were completed using Coot (Emsley et al, 2010) and Phenix.refine (Afonine et al, 2015; Liebschner et al, 2019). Data collection, processing, and refinement statistics are shown in Table 1. All structural figures were prepared using UCSF ChimeraX (Pettersen et al, 2021).

## Data Availability

Atomic structures and diffraction data have been deposited in the PDB with accession codes 9EGV (HOIL-1 RING2/Ub) and 9EGW (HOIL-1 RING2/Ub-maltose). All other data supporting the conclusions are available in the article, supplementary material, and source data.

## Supplementary Information

## Acknowledgements

We thank other members of the Lechtenberg and Komander laboratories for helpful discussions and suggestions, Geoffrey Kong (Monash Macromolecular Crystallisation Facility) for help with crystallisation, and Vineet Vaibhav for help with intact mass spectrometry. This work was funded by WEHI and National Health and Medical Research Council (NHMRC) Ideas Grants 1182757 (to BC Lechtenberg) and 2027601 (to ED Goddard-Borger); NHMRC Investigator Grants 1195038 (to J Silke), 1174941 (to RJ Payne), 2033340 (to ED Goddard-Borger), 1178122 (to D Komander), and 2016268 (to BC Lechtenberg); a Rebecca Cooper Fellowship (to ED Goddard-Borger); and Australian Research Council DECRA Fellowship DE230100634 (to L Corcilius). XS Wang is supported by an Australian Government Research Training Program Scholarship. The mass spectrometry analysis was performed at the WEHI Proteomics Facility. This research was undertaken in part using the MX2 beamline at the Australian Synchrotron, part of the Australian Nuclear Science and Technology Organisation (ANSTO), and made use of the Australian Cancer Research Foundation (ACRF) detector.

### Author Contributions

XS Wang: conceptualisation, resources, data curation, formal analysis, validation, investigation, visualisation, methodology, and writing—original draft, review, and editing.
J Jiou: validation, investigation, visualisation, methodology, and writing—review and editing.
A Cerra: data curation, validation, investigation, visualisation, methodology, and writing—review and editing.
SA Cobbold: conceptualisation, data curation, formal analysis, investigation, methodology, and writing—review and editing.
M Jochem: conceptualisation, data curation, formal analysis, investigation, methodology, and writing—review and editing.
KHT Mak: resources, investigation, methodology, and writing—review and editing.
L Corcilius: resources, funding acquisition, investigation, methodology, and writing—review and editing.
J Silke: conceptualisation, resources, supervision, funding acquisition, methodology, and writing—review and editing.
RJ Payne: conceptualisation, resources, supervision, funding acquisition, methodology, and writing—review and editing.
ED Goddard-Borger: conceptualisation, resources, funding acquisition, visualisation, methodology, and writing—review and editing.
D Komander: conceptualisation, resources, supervision, funding acquisition, methodology, and writing—review and editing.
BC Lechtenberg: conceptualisation, resources, data curation, formal analysis, supervision, funding acquisition, validation, investigation, visualisation, methodology, project administration, and writing—original draft, review, and editing.

**Conflict of Interest Statement**

The authors declare that they have no conflict of interest.

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
