## [Reviewer comments · Life Science Alliance]

Life Science Alliance

The RBR E3 ubiquitin ligase HOIL-1 can ubiquitinate diverse non-protein substrates in vitro

Xiangyi Wang, Jenny Jiou, Anthony Cerra, Simon Cobbold, Marco Jochem, Ka Hin Mak, Leo Corcilius, John Silke, Richard Payne, Ethan Goddard-Borger, David Komander, and Bernhard Lechtenberg

DOI: <https://doi.org/10.26508/lsa.202503243>

Corresponding author(s): *Bernhard Lechtenberg, Walter and Eliza Hall Institute of Medical Research*

Review Timeline:

Submission Date:	2025-01-31
Editorial Decision:	2025-01-31
Revision Received:	2025-03-20
Editorial Decision:	2025-03-21
Revision Received:	2025-03-22
Accepted:	2025-03-24

Transaction Report:

Please note that the manuscript was previously reviewed at another journal and the reports were taken into account in the decision-making process at *Life Science Alliance*.

Reviews

Referee #1 Review

Report for Author:

The manuscript by Wang et al explores the substrate specificity of HOIL-1. In particular, it examines the ability of HOIL-1 to ubiquitylate saccharides. The manuscript includes a mix of mechanistic insight and tool development. The tools will likely be of use to the field as it develops but the manuscript is a bit disconnected and the authors may wish to separate out the development tools to publish these separately from the structure/function analysis. There are also several points about the assays/data presentation and analysis that need to be addressed.

Main points:

- i) The manuscript relies on assays that distinguish modification of ubiquitin with small mass compounds (peptides and sugars). For many assays, modification of ubiquitin is monitored using SDS-PAGE and Coomassie staining, that is then followed by quantification. A number of these gels are difficult to assess, and it would be helpful if a labelled peptide was utilised to allow quantification as presented in Figure 1B. In the absence of labelled peptide quantification seems difficult? Perhaps a different approach to SDS-PAGE might be useful, with use of gels that have a greater ability to separate low molecular weight proteins?
- ii) In Figure 3 conclusions are made about the ability of HOIL-1 to ubiquitylate different disaccharides and this is compared to its ability to modify peptides. The graphs presented should include error bars and the conclusions should be revisited. I am not convinced that the activity differs between the different substrates? It also seems likely that the data in Figure 1A is repeated in Figure 3A, or is that a different data set? It maybe helpful to review data order/layout.

Minor points:

- i) Line 55, revise 'LUBAC components, components'
- ii) Line 58, to improve clarity delete 'to ubiquitin as a protein post-translational modification'.
- iii) Line 67/68, revise text.
- iv) Figure 1A and C should be combined and all included on the graph - if it is included.
- v) Fig 2 and sup Fig 2. Review colour scheme, the blue green overlay is difficult to see. Grey for one molecule may be helpful.
- vi) Figure 2D, label Q455 - N461.
- vii) Figure 5C/D should be included with Figure 4 as supplementary data. Generation of the fusion protein does not add to Figure 5.
- viii) Figure 5B states that different concentrations of the DUBs were used based on prior optimisation. However, it is not clear that this is relevant as the substrate is different and assays that use the same molar concentration of DUB should be included.

Referee #2 Review

Report for Author:

This study investigates carbohydrate ubiquitination by HOIL-1, a ubiquitin ligase which has recently been shown to conjugate Ub to sugar molecules under in vitro conditions. Non-proteinaceous Ub conjugation is an emerging field in Ub biology and the manuscript by Wang et al. is a welcome addition to this research area. A major aim of this work is to systematically characterise carbohydrate

substrate specificity. The authors deploy an in vitro assay which shows that the HOIL-1 mediated ubiquitination activity of different mono- and di-saccharides are all very similar, suggesting that the modification of sugar molecules is a rather unspecific event. The authors solve a crystal structure of a complex between HOIL-1 and maltose modified Ub, but unfortunately the structure is of limited value as no density for the sugar moiety could be observed. The authors claim that their HOIL-1 structure without maltose linked Ub reveal a rearrangement of His510 which undergoes a side chain rotation between an inactive and active conformation. The role of His510, which is known to act as catalytic base, is further investigated in ubiquitination assays with different peptides as model substrates to explore HOIL-1 preference to generate isopeptide bonds to Lys residues vs. oxyester bonds to Ser/Thr residues. Here the authors conclude that the catalytic His510 is required for ubiquitination of hydroxy groups and at the same time restricts amide linked ubiquitination which can only be observed in the absence of His510. Another focus of this manuscript is the development of a protocol for large-scale preparation of ubiquitinated saccharides by HOIL-1 with the aim to deploy these carbohydrate-Ub's as model substrates. A set of deubiquitinases was used to test for specific Ub-maltose cleavage which shows that all DUBs with the exception of OTULIN are capable to hydrolyse Ub-maltose. This is one of the first studies which investigates the biochemistry of non-protein ubiquitination in detail. The manuscript contains some interesting data which can be used as useful resource for ubiquitin labs working on this topic. However, the results presented here raise several questions (see below) and the paper does not really provide new answers how HOIL-1 modifies carbohydrates. The role of His510 and its importance for the formation of oxyester linkages with sugars and protein (ubiquitin) was already shown in several papers (Xu et al., *Sci Adv*, 2023; Wu et al., *Front Mol Biosci*, 2023; Wang et al., *Nat Commun*, 2023). Overall, this manuscript presents a good mix of solid data which are relevant for a specialist audience but might not be of great interest for a broader readership.

Major comments:

- 1) Fig 1 D-E show peptide ubiquitination by HOIL-1 in context of wt and H510A. While autoubiquitination of wt HOIL is oxyester-linked, the mutant H510A displays bands of isopeptide- but also oxyester-linked autoubiquitinated HOIL. The authors claim that H510 is required to deprotonate the hydroxy group for the formation of O-linked ubiquitination. If that is the case, then no oxyester linked autoubiquitination bands should be observed for HOIL-1 H510A. Can the authors explain this discrepancy?
- 2) Experiments in Fig 1A-C were carried out with full length HOIL-1 but experiments in Fig1 D-E were performed with a smaller construct comprising the RBR only. Why? The authors should explain why they have selected truncated HOIL-1 for the experiment in panel D-E. Does full length and truncated HOIL-1 have the same activity? For example, the HOIL-1 NZF domain which is upstream of the RBR binds with high affinity to M1-linked Ub chains. Since M1-Ub2 is required for HOIL-1 activation, does the absence/presence of NZF have an effect on HOIL ubiquitination activity and/or autoubiquitination?
- 3) The authors discuss structural differences observed between the free RING2 domain and the Ub bound RING2 domain. They suggest that the Ub bound RING2 structure adopts an active conformation leading to rearrangement of His510 and other residues in this area. To substantiate these findings, the authors should compare their structure with other HOIL-1 RING2 structures recently solved without ubiquitin (PDB ID: 8BVL, 7YUJ). Are these structures have the same inactive conformation?
- 4) The authors claim that H510 "is a critical switch to fine-tune HOIL-1 activity". This statement is somewhat misleading as it suggests that H510 has a regulatory function to switch between oxyester and isopeptide bond formation. However, this is only the case when H510 is artificially removed in an in vitro experiment which has no relevance in a cellular context where this mutation does not exist.

Minor comment:

Page 12; 378: "...discharge ubiquitin onto a solvent molecule (water/hydroxide ion) ..." sounds strange. Hydrolysis of the thioester would be better.

Referee #3 Review

Report for Author:

The manuscript provides new and interesting insights into HOIL-1 substrate specificity, and the ubiquitination of non-proteinaceous substrates. The authors use their structural data to show how the catalytic H510 enables O-linked ubiquitination and prevents lysine ubiquitination. They also present a method for the efficient production of pure ubiquitinated saccharides. This is relevant for the scientific community to enable the quantitative study of these processes as current tools for studying non-proteinaceous ubiquitination are limited. As a proof of concept, the author tested different DUBs using ubiquitin-maltose as a substrate.

The authors propose that HOIL-1 may ubiquitinate saccharides in cellular environments when present in high local concentrations (e.g. maltose in glycogen, Kdo on bacterial LPS, MurNAc on peptidoglycans). Producing these non-proteinaceous ubiquitin conjugates may be useful in future investigations, especially in the context of ubiquitinating intracellular pathogens (which falls in line with LUBAC's role in immune signaling).

A minor concern is their secondary crystallization result with Ub-maltose/HOIL-RING2. The authors state that they were only capable of observing electron density for the C-terminus of the Ub but not the maltose. They chalk it up to possible reduced binding affinity between the RING2 and Ub-maltose. It would be nice if they could provide additional information to confirm Ub-maltose in the crystal (e.g., running the crystals on a gel or mass spec). While there is little doubt that the maltose-Ub species is stable in the crystal drop, the crystallization process could have selected unmodified Ub.

Other minor points

- While the constitutively active HOIL-1 mutant fits into the story, I would have expected that they use it for the large-scale production of the His-Ub-maltose. I noticed that in the abstract, they even say: "Finally, we utilise HOIL-1's in vitro non-proteinaceous ubiquitination activity and an engineered, constitutively active HOIL-1 variant to produce preparative amounts of different ubiquitinated saccharides that can be used as tool compounds (lines 29-32)". Some explanation for why they do not actually use the mutant would be helpful.
- For Figure 1, "Experiments in panels D and E were performed at least twice (except for the right gel in panel E) with consistent results (ll. 855-856)". It would have been nice to have all of them as duplicates.
- For figure 2, could the elements described in ll. 180-183 be shown more clearly, or would make it too crowded?
- "The slower migrating band disappears upon treatment with 134 hydroxylamine (NH₂OH) which specifically breaks oxyester bonds while leaving (iso)-peptide bonds intact" (lines 133-135). From my understanding, hydroxylamine also breaks thioester bonds while "specifically" suggests that it is only oxyester bonds.
- Lines 193-194 state: "Therefore, after establishing that HOIL-1 efficiently ubiquitinates hydroxyl groups in Ser-residues over Lys residues..." My understanding from the results is that the modification of Thr is less efficient than the Ser ubiquitination, but still more efficient than the Lys ubiquitination, so

maybe the Thr residues should be added here?

- "We used the same crystallisation conditions as described for HOIL-1 RING2/ubiquitin above" (lines 293-294), I am not quite sure what the authors mean and which crystallization conditions they are referring to? In the manuscript, the methods are at the end.
- In Supplementary figure 5 the first lane lacks a label and I am not sure if I understand what the additional band in the +OTUD1 condition is. It is not labeled either.
- In supplementary figure 6, the elution is called "E2" but there doesn't seem to be an E1.

Language/grammar/typos:

- ll. 365-367: "Our crystal structure of the HOIL-1 RING2 domain with and without ubiquitin shows that His510 forms part a dynamic HOIL-1 catalytic centre that includes the loop containing the catalytic Cys460 and Trp462." I think it should read "forms part of a" or "is part of a"
- ll. 464-467: "The His-3C-M1-di-Ub(G76V)-HOIL-1 fusion protein, was purified basically as described above for full-length HOIL-1 but with an additional wash step with 30 mM imidazole in high salt purification buffer before elution in high salt purification buffer supplemented with 300mM imidazole and the His-tag was not cleaved." I do not understand the comma in this sentence.
- ll. 482-484: "Reactions in Figure 2 were further treated with 1.5 M NH₂OH (Sigma-Aldrich 467804) as indicated for 30 min at 30{degree sign}C before addition of 1x LDS sample buffer supplemented with 0.1 M DTT." There are only structures in figure 2, no reactions, this is probably referring to figure 1D/E?
- In the method section, they should be consistent whether to include a space between the pH and the following number (pH 7 vs. pH6, both are used).
- For large scale/large-scale) preparation, they are not consistent with the hyphenation (e.g. ll. 893 and 895).
- In the method section, they always state that 1x LDS buffer was used except for line 535. If this was different, it should be explained, but I assume they just forgot to add the 1x.
- Supplementary figure legend 1, l. 925 should read "contoured to 1 σ ", not 1s, especially because they use the greek letter themselves later in supplementary figure legend 4.

January 31, 2025

Re: Life Science Alliance manuscript #LSA-2025-03243-T

Dr. Bernhard C. Lechtenberg
Walter and Eliza Hall Institute
Ubiquitin Signalling Division
1G Royal Parade
Parkville, VIC 3052
Australia

Dear Dr. Lechtenberg,

Thank you for submitting your manuscript entitled "The RBR E3 ubiquitin ligase HOIL-1 can ubiquitinate diverse non-proteinaceous substrates in vitro" to Life Science Alliance. We invite you to submit a revised manuscript addressing the Reviewer comments.

Thank you for this interesting contribution to Life Science Alliance. We are looking forward to receiving your revised manuscript.

Sincerely,

Eric Sawey, PhD
Executive Editor
Life Science Alliance
<http://www.lsa-journal.org>

B. MANUSCRIPT ORGANIZATION AND FORMATTING:

We thank the reviewers for their insightful comments and suggestions to improve our manuscript. We respond to each individual reviewers' comments below with reviewers' comments in black font and our response in blue font.

Referee #1:

The manuscript by Wang et al explores the substrate specificity of HOIL-1. In particular, it examines the ability of HOIL-1 to ubiquitylate saccharides. The manuscript includes a mix of mechanistic insight and tool development. The tools will likely be of use to the field as it develops but the manuscript is a bit disconnected and the authors may wish to separate out the development tools to publish these separately from the structure/function analysis. There are also several points about the assays/data presentation and analysis that need to be addressed.

RESPONSE: We thank the reviewer for their comments. Our manuscript provides insights on the HOIL-1 mechanism and tool development for the non-proteinaceous ubiquitin field. In our view, these two parts of the manuscript build on each other, as the tool development resulted immediately from our biochemical observations that HOIL-1 can ubiquitinate a vast variety of non-proteinaceous substrates. We therefore prefer to publish these aspects of our work in this single manuscript.

Main points:

i) The manuscript relies on assays that distinguish modification of ubiquitin with small mass compounds (peptides and sugars). For many assays, modification of ubiquitin is monitored using SDS-PAGE and Coomassie staining, that is then followed by quantification. A number of these gels are difficult to assess, and it would be helpful if a labelled peptide was utilised to allow quantification as presented in Figure 1B. In the absence of labelled peptide quantification seems difficult? Perhaps a different approach to SDS-PAGE might be useful, with use of gels that have a greater ability to separate low molecular weight proteins?

RESPONSE: We repeated all experiments 2-3x as indicated in the figure legends with, in our view, very high reproducibility between independent repeats. We have however now removed Fig 1B as similar data are presented in Fig 3A. Nevertheless, we believe that these data highlight that the gel approach is suitable for detecting the ubiquitination of peptides. Importantly, we observe a clear gap between the bands of free ubiquitin and ubiquitin conjugated to the peptides, enabling accurate quantification. We had previously tested multiple gel systems and selected the system (1mm 12% NuPAGE Bis-Tris Mini Protein Gels, Invitrogen) that provided the best separation between the bands. We now specify the gels used in the Methods section.

We had previously tested the labelled peptide approach (using FITC-labelled peptides) also suggested by the reviewer but ran into technical issues that we were unable to resolve. The main issue was that the fluorescent scan was dominated by the signal from the unmodified peptide due to the high substrate/peptide concentrations in our assay. Further, we observed that the unmodified FITC-labelled peptide migrated much slower on the SDS-PAGE gel than expected, presumably due to a large effect of the FITC label on migration of the small 5-mer peptide, thereby further complicating detection of a potentially formed Ub-peptide-FITC species.

We acknowledge the general difficulty with the SDS-PAGE-based readout for smaller substrates. For this reason, we utilise MALDI-MS to better quantify the ubiquitinated mono-saccharide substrates in the later parts of our manuscript.

ii) In Figure 3 conclusions are made about the ability of HOIL-1 to ubiquitylate different disaccharides and this is compared to its ability to modify peptides. The graphs presented should include error bars and the conclusions should be revisited. I am not convinced that the activity differs between the different substrates? It also seems likely that the data in Figure 1A is repeated in Figure 3A, or is that a different data set? It maybe helpful to review data order/layout.

RESPONSE: Our graphs include all data points as recommended by the journal for small number of repeats (<https://www.life-science-alliance.org/manuscript-prep#stats>). The connecting lines in our graphs connect the mean of the datapoints. We agree with the journal's recommendation that this representation provides the most accurate overview of our data. Nevertheless, we investigated additionally including the mean with error bars (SEM) as suggested by the reviewer with two examples shown in Figure for reviewers 1. In our view, the addition of error bars in scatter plots (e.g. in Fig 3C, D) does not add much additional information and makes the graphs more difficult to view as they get very crowded. However, if the reviewer insists, we can change the relevant graphs. Note that we include all data points, mean and error bars in the less crowded bar graph in new Fig S4D.

We agree with the reviewer that there is no clear difference in activity with different disaccharide substrates, and our conclusion for Fig. 3C/D is (p. 9, ll. 247-249): "In summary, HOIL-1 can ubiquitinate all disaccharides tested with only marginal differences in the rate of ubiquitination, suggesting that HOIL-1 has broad substrate specificity for different model disaccharides".

Data in Figure 1A and Figure 3A are from different experiments with different time-points. Figure 1A contains time points 0, 10, 20, 30 and 60 min, whereas Figure 3A contains time points 0, 2.5, 5, 10 and 30 min.

Figure for reviewers 1: Comparison of scatter plots with individual data points as currently shown in our manuscript (left) with scatter plots additionally showing the mean and error bars (SEM) (right).

Minor points:

i) Line 55, revise 'LUBAC components, components'

RESPONSE: We have rephrased this sentence.

ii) Line 58, to improve clarity delete 'to ubiquitin as a protein post-translational modification'.

RESPONSE: We have rephrased this sentence.

iii) Line 67/68, revise text.

RESPONSE: We have rephrased this sentence.

iv) Figure 1A and C should be combined and all included on the graph - if it is included.

RESPONSE: Given that the experiments in Fig 1C do not show any ubiquitin conjugation to either of the two peptides, we do not think a graph is necessary to highlight this. Further, we show quantification of ubiquitination of Ser- and Thr-peptides alongside maltose from a different experiment in Fig 2A. Therefore, we now removed the graph in Fig 1C to simplify the figure.

v) Fig 2 and sup Fig 2. Review colour scheme, the blue green overlay is difficult to see. Grey for one molecule may be helpful.

RESPONSE: We changed the colour of the free HOIL-1 RING2 molecule in Fig 2 and Fig S1 from blue to grey as suggested by the reviewer. (Please note that Fig S2 indicated by the reviewer does not include structural figures, and we concluded that this was a typo, and the reviewer was meant to refer to Fig S1. Please correct if we misinterpreted).

vi) Figure 2D, label Q455 - N461.

RESPONSE: We now more clearly define the region as the active site loop in the main text and have labelled this loop in Figure 2D. We also include an additional label for the extended β -strand.

vii) Figure 5C/D should be included with Figure 4 as supplementary data. Generation of the fusion protein does not add to Figure 5.

RESPONSE: We agree with the reviewer that Fig 5C/D was disconnected from Fig 5A/B. We now moved Figure 5 C/D to a new Fig 6.

viii) Figure 5B states that different concentrations of the DUBs were used based on prior optimisation. However, it is not clear that this is relevant as the substrate is different and assays that use the same molar concentration of DUB should be included.

RESPONSE: We agree with the reviewer that the enzyme concentrations are an important consideration in this assay. We decided to use the DUBs at the concentrations that have previously been identified to show robust cleavage of true substrates (and verified in our hands in Fig S5) while avoiding non-specific cleavage of suboptimal substrates, i.e., other types of ubiquitin linkages. While we agree with the reviewer that it may be difficult to transfer these insights to the cleavage of ubiquitin-maltose, we think this approach is still warranted in our experiments that we describe as “proof-of-principle” experiments (p. 12, l. 395). To fully understand the specificity and activity of the different DUBs against non-proteinaceous substrates, more sophisticated experiments are required in which the optimal concentration for each DUB would need to be tested. We believe that this is beyond the scope of the current manuscript. We now added a sentence to the manuscript to acknowledge this: “Our observations may form the basis to investigate DUB activity and specificity for non-proteinaceous substrates more thoroughly.” (p. 12, l. 400-402).

Referee #2:

This study investigates carbohydrate ubiquitination by HOIL-1, a ubiquitin ligase which has recently been shown to conjugate Ub to sugar molecules under in vitro conditions. Non-proteinaceous Ub conjugation is an emerging field in Ub biology and the manuscript by Wang et al. is a welcome addition to this research area. A major aim of this work is to systematically characterise carbohydrate substrate specificity. The authors deploy an in vitro assay which shows that the HOIL-1 mediated ubiquitination activity of different mono- and di-saccharides are all very similar, suggesting that the modification of sugar molecules is a rather unspecific event. The authors solve a crystal structure of a complex between HOIL-1 and maltose modified Ub, but unfortunately the structure is of limited value as no density for the sugar moiety could be observed. The authors claim that their HOIL-1 structure without maltose linked Ub reveal a rearrangement of His510 which undergoes a side chain rotation between an inactive and active conformation. The role of His510, which is known to act as catalytic base, is further investigated in ubiquitination assays with different peptides as model substrates to explore HOIL-1 preference to generate isopeptide bonds to Lys residues vs. oxyester bonds to Ser/Thr residues. Here the authors conclude that the catalytic His510 is required for ubiquitination of hydroxy groups and at the same time restricts amide linked ubiquitination which can only be observed in the absence of His510. Another focus of this manuscript is the development of a protocol for large-scale preparation of ubiquitinated saccharides by HOIL-1 with the aim to deploy these carbohydrate-Ub's as model substrates. A set of deubiquitinases was used to test for specific Ub-maltose cleavage which shows that all DUBs except for OTULIN are capable to hydrolyse Ub-maltose. This is one of the first studies which investigates the biochemistry of non-protein ubiquitination in detail. The manuscript contains some interesting data which can be used as useful resource for ubiquitin labs working on this topic. However, the results presented here raise several questions (see below) and the paper does not really provide new answers how HOIL-1 modifies carbohydrates. The role of His510 and its importance for the formation of oxyester linkages with sugars and protein (ubiquitin) was already shown in several papers (Xu et al., Sci Adv, 2023; Wu et al., Front Mol Biosci, 2023; Wang et al., Nat Commun, 2023). Overall, this manuscript presents a good mix of solid data which are relevant for a specialist audience but might not be of great interest for a broader readership.

Major comments:

1) Fig 1 D-E show peptide ubiquitination by HOIL-1 in context of wt and H510A. While autoubiquitination of wt HOIL is oxyester-linked, the mutant H510A displays bands of isopeptide- but also oxyester-linked autoubiquitinated HOIL. The authors claim that H510 is required to deprotonate the hydroxy group for the formation of O-linked ubiquitination. If that is the case, then no oxyester linked autoubiquitination bands should be observed for HOIL-1 H510A. Can the authors explain this discrepancy?

RESPONSE: The reviewer makes a valid point that we had not adequately described in our initial manuscript. The ubiquitination of a substrate is dependent on the reactivity of the accepting group as well as the residence time of that acceptor in the active site. The deprotonation mentioned by the reviewer affects the reactivity, and this is dependent on the chemical environment of the accepting group.

Therefore, one reason for the observed oxyester-linked autoubiquitination of HOIL-1 H510A may be a solvent-exposed acceptor Ser or Thr residue in HOIL-1 with a higher propensity of deprotonation due to the local chemical environment. Regarding the residency time, a Ser/Thr residue close to the HOIL-1 active site may be ubiquitinated due to its presence in the active site even though it is inefficient. We now include a brief discussion of these possibilities in our manuscript (p. 14, ll. 419–425)

2) Experiments in Fig 1A-C were carried out with full length HOIL-1 but experiments in Fig1 D-E were performed with a smaller construct comprising the RBR only. Why? The authors should explain why they have selected truncated HOIL-1 for the experiment in panel D-E. Does full length and truncated HOIL-1 have the same activity? For example, the HOIL-1 NZF domain which is upstream of the RBR binds with high affinity to M1-linked Ub chains. Since M1-Ub2 is required for HOIL-1 activation, does the absence/presence of NZF have an effect on HOIL ubiquitination activity and/or autoubiquitination?

RESPONSE: The experiments in previous Figure 1 D,E (now Fig 1C,D) are based on our previous work (Wang et al, Nat Comms 2023; PMID: 36631489). In that study we generated the His510Ala mutant in the helix-RBR background that we now also used in Fig 1C,D. We understand the reviewer's concerns. However, in that previous publication we showed that the NZF domain does not significantly affect activation of HOIL-1 by di-ubiquitin. Specifically, in Fig 1d of Wang et al, Nat Comms 2023, we show that the EC50 of di-ubiquitin activation between a HOIL helix-RBR construct (as used in the current manuscript) is similar to the EC50 determined with a construct containing the NZF. Importantly, the same results were reported in publications by two other laboratories: Kellsall et al EMBO J 2022 (PMID: 35274759) and Xu et al Sci Adv 2023 (PMID: 37831767). In addition, it is important to note that the experiments in Fig 1C,D are internally controlled by comparing the activities of WT HOIL-1 with HOIL-1 H510A in the same HOIL-1 background. In summary, we believe that using the shorter constructs in these experiments does not affect the conclusions. We now more clearly describe that we use a different construct for these experiments compared to the other experiments (p. 6, ll. 150–152).

3) The authors discuss structural differences observed between the free RING2 domain and the Ub bound RING2 domain. They suggest that the Ub bound RING2 structure adopts an active conformation leading to rearrangement of His510 and other residues in this area. To substantiate these findings, the authors should compare their structure with other HOIL-1 RING2 structures recently solved without ubiquitin (PDB ID: 8BVL, 7YUJ). Are these structures have the same inactive conformation?

RESPONSE: We thank the reviewer for this suggestion. The other unbound HOIL-1 RING2 structures reported in PDB IDs 7YUJ and 8BVL show the same conformation observed in our unbound HOIL-1. We now include an overlay of these three structures in new Figure S1E and briefly describe the analysis in the main text (p. 6-7, ll. 236-245).

4) The authors claim that H510 "is a critical switch to fine-tune HOIL-1 activity". This statement is somewhat misleading as it suggests that H510 has a regulatory function to switch between oxyester and isopeptide bond formation. However, this is only the case when H510 is artificially removed in an in vitro experiment which has no

relevance in a cellular context where this mutation does not exist.

RESPONSE: We agree that the word “switch” was not accurate. We now changed this sentence to: “His510 therefore is a critical element to fine-tune HOIL-1 activity towards oxyester linked ubiquitination.” (p. 6, ll. 174-176).

Minor comment:

Page 12; 378: "...discharge ubiquitin onto a solvent molecule (water/hydroxide ion) ..." sounds strange. Hydrolysis of the thioester would be better.

RESPONSE: We have rephrased this sentence.

Referee #3:

The manuscript provides new and interesting insights into HOIL-1 substrate specificity, and the ubiquitination of non-proteinaceous substrates. The authors use their structural data to show how the catalytic H510 enables O-linked ubiquitination and prevents lysine ubiquitination. They also present a method for the efficient production of pure ubiquitinated saccharides. This is relevant for the scientific community to enable the quantitative study of these processes as current tools for studying non-proteinaceous ubiquitination are limited. As a proof of concept, the author tested different DUBs using ubiquitin-maltose as a substrate.

The authors propose that HOIL-1 may ubiquitinate saccharides in cellular environments when present in high local concentrations (e.g. maltose in glycogen, Kdo on bacterial LPS, MurNAc on peptidoglycans). Producing these non-proteinaceous ubiquitin conjugates may be useful in future investigations, especially in the context of ubiquitinating intracellular pathogens (which falls in line with LUBAC's role in immune signaling).

A minor concern is their secondary crystallization result with Ub-maltose/HOIL-RING2. The authors state that they were only capable of observing electron density for the C-terminus of the Ub but not the maltose. They chalk it up to possible reduced binding affinity between the RING2 and Ub-maltose. It would be nice if they could provide additional information to confirm Ub-maltose in the crystal (e.g., running the crystals on a gel or mass spec). While there is little doubt that the maltose-Ub species is stable in the crystal drop, the crystallization process could have selected unmodified Ub.

RESPONSE: We collected the diffraction datasets from these crystals in 2023, and therefore did not have any crystals left to perform this analysis. However, we still have the HOIL-1/Ub-maltose complex available and used that to grow new HOIL-1/Ub-maltose crystals. We separately harvested and redissolved multiple crystals from two different crystallisation trays and applied these solutions to intact mass spectrometry. From these samples, we were able to detect ubiquitin-maltose (~75% of the total peak area in both experiments) as well as a smaller fraction of unmodified ubiquitin (~20% of the total peak area), confirming that most of the ubiquitin in the crystals is still conjugated to maltose. We also tested the HOIL-1/Ub-maltose solution used to grow these crystals. Here, nearly all the ubiquitin is conjugated to maltose, with free ubiquitin basically undetectable as expected from our intact MS performed on the Ub-maltose preparation (shown in Fig 4E) used to generate the HOIL-1/Ub-maltose complex. These observations indicated that the unconjugated ubiquitin observed in the crystals must have derived from ubiquitin-maltose either during crystallisation or sample processing between crystallisation and mass spectrometry. Nevertheless, we conclude that the vast majority of the ubiquitin in our structure is conjugated to maltose and the lack of clear density for maltose likely indicates that the maltose does not bind the HOIL-1 RING2 in a defined orientation. As these intact MS experiments were performed on crystals from a different crystallisation experiment as the structures reported in our manuscript, we also determined the structure of HOIL-1/Ub-maltose using one of the crystals taken from the same crystallisation tray as one of the mass spec samples. As in our previous structure, we do not observe interpretable density for the maltose moiety, and we therefore believe that these MS results are also representative of our previous

crystals used to determine the reported structures. We now include the intact MS analysis of our new crystal samples in Fig S4D of our manuscript, included a short description in the main text (p. 11, ll. 328-329) and updated the methods section (p. 20, ll. 623-627).

Other minor points

- While the constitutively active HOIL-1 mutant fits into the story, I would have expected that they use it for the large-scale production of the His-Ub-maltose. I noticed that in the abstract, they even say: "Finally, we utilise HOIL-1's in vitro non-proteinaceous ubiquitination activity and an engineered, constitutively active HOIL-1 variant to produce preparative amounts of different ubiquitinated saccharides that can be used as tool compounds (lines 29-32)". Some explanation for why they do not actually use the mutant would be helpful.

RESPONSE: We generated the constitutively active HOIL-1 fusion only after we generated a large batch of the His-Ub-maltose that we used for our subsequent experiments in this manuscript. The His-Ub-maltose used for the crystallography and DUB assays described in the manuscript was generated using full-length WT HOIL-1 with M1-linked di-Ub as an allosteric activator, as described in the Methods section under "Large scale ubiquitinated saccharide production and purification" (p. 19, ll. 593-605). We now reworded the sentence in the abstract to avoid any misunderstandings: "We utilise HOIL-1's in vitro non-proteinaceous ubiquitination activity to produce preparative amounts of different ubiquitinated saccharides that can be used as tool compounds and standards in the rapidly emerging field of non-proteinaceous ubiquitination. Finally, we report an engineered, constitutively active HOIL-1 variant that simplifies in vitro generation of ubiquitinated saccharides." (p. 2, ll. 36-41).

- For Figure 1, "Experiments in panels D and E were performed at least twice (except for the right gel in panel E) with consistent results (ll. 855-856)". It would have been nice to have all of them as duplicates.

RESPONSE: We now repeated the experiment reported in the right panel of Figure 1E (now Fig 1D). The repeat is consistent with the previously reported experiment. We include Fig 1D and the new repeat in Figure for reviewers 2:

Figure for reviewers 2: (Left) Reproduction of Figure 1D of the manuscript. (Right) Repeat of the right gel of Fig 1D showing consistent results between the two repeat experiments despite some issues with the gel quality in the repeat.

- For figure 2, could the elements described in ll. 180-183 be shown more clearly, or would make it too crowded?

RESPONSE: We now more clearly define the region as the active site loop in the main text and have labelled this loop in Figure 2D. We also include an additional label for the extended β -strand.

- "The slower migrating band disappears upon treatment with 134 hydroxylamine (NH₂OH) which specifically breaks oxyester bonds while leaving (iso)-peptide bonds intact" (lines 133-135). From my understanding, hydroxylamine also breaks thioester bonds while "specifically" suggests that it is only oxyester bonds.

RESPONSE: We removed "specifically".

- Lines 193-194 state: "Therefore, after establishing that HOIL-1 efficiently ubiquitinates hydroxyl groups in Ser-residues over Lys residues..." My understanding from the results is that the modification of Thr is less efficient than the Ser ubiquitination, but still more efficient than the Lys ubiquitination, so maybe the Thr residues should be added here?

RESPONSE: We have reworded this statement to: "Therefore, after establishing that HOIL-1 most efficiently ubiquitinates hydroxyl groups in Ser residues over Thr residues and does not ubiquitinate Lys residues, ...".

- "We used the same crystallisation conditions as described for HOIL-1 RING2/ubiquitin above" (lines 293-294), I am not quite sure what the authors mean and which crystallization conditions they are referring to? In the manuscript, the methods are at the end.

RESPONSE: We rephrased this part to: "We crystallised HOIL-1 RING/ubiquitin-maltose under the same conditions as HOIL-1 RING2/ubiquitin...".

- In Supplementary figure 5 the first lane lacks a label and I am not sure if I understand what the additional band in the +OTUD1 condition is. It is not labeled either.

RESPONSE: The first lane is the sample without DUB treatment. The additional band in the +OTUD1 condition is K63 tri-ubiquitin, one of the intermediate cleavage products from K63 tetra-ubiquitin cleavage by OTUD1. We now added labels to indicate this.

- In supplementary figure 6, the elution is called "E2" but there doesn't seem to be an E1.

RESPONSE: We revised the labels and figure legend for this figure to better describe the different samples visualised on the SDS-PAGE.

Language/grammar/typos:

- ll. 365-367: "Our crystal structure of the HOIL-1 RING2 domain with and without ubiquitin shows that His510 forms part a dynamic HOIL-1 catalytic centre that includes the loop containing the catalytic Cys460 and Trp462." I think it should read "forms part of a" or "is part of a"

- ll. 464-467: "The His-3C-M1-di-Ub(G76V)-HOIL-1 fusion protein, was purified basically as described above for full-length HOIL-1 but with an additional wash step with 30 mM imidazole in high salt purification buffer before elution in high salt purification buffer supplemented with 300mM imidazole and the His-tag was not cleaved." I do not understand the comma in this sentence.

- ll. 482-484: "Reactions in Figure 2 were further treated with 1.5 M NH₂OH (Sigma-

Aldrich 467804) as indicated for 30 min at 30{degree sign}C before addition of 1x LDS sample buffer supplemented with 0.1 M DTT." There are only structures in figure 2, no reactions, this is probably referring to figure 1D/E?

- In the method section, they should be consistent whether to include a space between the pH and the following number (pH 7 vs. pH6, both are used).
- For large scale/large-scale) preparation, they are not consistent with the hyphenation (e.g. ll. 893 and 895).
- In the method section, they always state that 1x LDS buffer was used except for line 535. If this was different, it should be explained, but I assume they just forgot to add the 1x.
- Supplementary figure legend 1, l. 925 should read "contoured to 1σ ", not 1s, especially because they use the greek letter themselves later in supplementary figure legend 4.

RESPONSE: We have revised these sections of our manuscript according to the reviewer's suggestions.

In addition to the points raised by the reviewers, we also identified an inaccuracy with the HOIL-1 RING2 construct used for crystallisation. The correct boundaries for this construct used are residues 425–510 rather than 412–510 as incorrectly reported in our initial submission. We revised our manuscript and are also updating our PDB entries accordingly. Finally, we corrected a few typos and inconsistencies in the text.

March 21, 2025

RE: Life Science Alliance Manuscript #LSA-2025-03243-TR

Dr. Bernhard C. Lechtenberg
Walter and Eliza Hall Institute of Medical Research
Ubiquitin Signalling Division
1G Royal Parade
Parkville, VIC 3052
Australia

Dear Dr. Lechtenberg,

Thank you for submitting your revised manuscript entitled "The RBR E3 ubiquitin ligase HOIL-1 can ubiquitinate diverse non-proteinaceous substrates in vitro". We would be happy to publish your paper in Life Science Alliance pending final revisions necessary to meet our formatting guidelines.

- please be sure that the authorship listing and order is correct
- please add the X and Bluesky handles of your host institute/organization as well as your own or/and one of the authors in our system
- please note that titles in the system and manuscript file must match
- please add callouts for Figures S3A-C and S6B to your main manuscript text

FIGURE CHECK:

- please indicate what the dashed vertical lines are for in Figures 1 and S2A. Sometimes these are used to indicate blot splicing, which does not appear to be the case here. If just to improve figure readability, then please state this in the relevant figure legends.

LSA now encourages authors to provide a 30-60 second video where the study is briefly explained. We will use these videos on social media to promote the published paper and the presenting author (for examples, see <https://docs.google.com/document/d/1-UWCfbE4pGcDdcgzcmiuJl2XMBJnxKYeqRvLLrLS08s/edit?usp=sharing>). Corresponding or first-authors are welcome to submit the video. Please submit only one video per manuscript. The video can be emailed to contact@life-science-alliance.org

A. FINAL FILES:

B. MANUSCRIPT ORGANIZATION AND FORMATTING:

Sincerely,

March 24, 2025

RE: Life Science Alliance Manuscript #LSA-2025-03243-TRR

Dr. Bernhard C. Lechtenberg
Walter and Eliza Hall Institute of Medical Research
Ubiquitin Signalling Division
1G Royal Parade
Parkville, VIC 3052
Australia

Dear Dr. Lechtenberg,

Thank you for submitting your Research Article entitled "The RBR E3 ubiquitin ligase HOIL-1 can ubiquitinate diverse non-protein substrates in vitro". It is a pleasure to let you know that your manuscript is now accepted for publication in Life Science Alliance. Congratulations on this interesting work.

DISTRIBUTION OF MATERIALS:

Again, congratulations on a very nice paper. I hope you found the review process to be constructive and are pleased with how the manuscript was handled editorially. We look forward to future exciting submissions from your lab.

Sincerely,
